# Correlative Information Maximization: A Biologically Plausible Approach to Supervised Deep Neural Networks without Weight Symmetry

**Bariscan Bozkurt**[1,2,3]    **Cengiz Pehlevan**[4,5]    **Alper T. Erdogan**[2,3]

[1] Gatsby Computational Neuroscience Unit, UCL, United Kingdom
[2]KUIS AI Center, Koc University, Turkey    [3]EEE Department, Koc University, Turkey
[4]John A. Paulson School of Engineering & Applied Sciences and Center for
Brain Science, Harvard University, Cambridge, 02138 MA, USA
[5]Kempner Institute for the Study of Natural and Artificial Intelligence
{bbozkurt15, alperdogan}@ku.edu.tr  cpehlevan@seas.harvard.edu

## Abstract

The backpropagation algorithm has experienced remarkable success in training large-scale artificial neural networks; however, its biological plausibility has been strongly criticized, and it remains an open question whether the brain employs supervised learning mechanisms akin to it. Here, we propose correlative information maximization between layer activations as an alternative normative approach to describe the signal propagation in biological neural networks in both forward and backward directions. This new framework addresses many concerns about the biological-plausibility of conventional artificial neural networks and the backpropagation algorithm. The coordinate descent-based optimization of the corresponding objective, combined with the mean square error loss function for fitting labeled supervision data, gives rise to a neural network structure that emulates a more biologically realistic network of multi-compartment pyramidal neurons with dendritic processing and lateral inhibitory neurons. Furthermore, our approach provides a natural resolution to the weight symmetry problem between forward and backward signal propagation paths, a significant critique against the plausibility of the conventional backpropagation algorithm. This is achieved by leveraging two alternative, yet equivalent forms of the correlative mutual information objective. These alternatives intrinsically lead to forward and backward prediction networks without weight symmetry issues, providing a compelling solution to this long-standing challenge.

## 1   Introduction

How biological neural networks learn in a supervised manner has long been an open problem. The backpropagation algorithm [1], with its remarkable success in training large-scale artificial neural networks and intuitive structure, has inspired proposals for how biologically plausible neural networks can perform the necessary efficient credit-assignment for supervised learning in deep neural architectures [2]. Nonetheless, certain aspects of the backpropagation algorithm, combined with the oversimplified nature of artificial neurons, have been viewed as impediments to proposals rooted in this inspiration [3].

One of the primary critiques regarding the biological plausibility of the backpropagation algorithm is the existence of a parallel backward path for backpropagating error from the output towards the input, which uses the same synaptic weights as the forward path [1, 2, 4]. Although such weight transport, or weight symmetry, is deemed highly unlikely based on experimental evidence [3, 4],

37th Conference on Neural Information Processing Systems (NeurIPS 2023).

some biologically plausible frameworks still exhibit this feature, which is justified by the symmetric structure of the Hebbian updates employed in these frameworks [2, 5, 6].

The concerns about the simplicity of artificial neurons have been addressed by models which incorporate multi-compartment neuron models into networked architectures and ascribe important functions to dendritic processing in credit assignment [7, 8, 9, 10]. This new perspective has enabled the development of neural networks with improved biological plausibility.

In this article, we propose the use of correlative information maximization (CorInfoMax) among consecutive layers of a neural network as a new supervised objective for biologically plausible models, which offers

- a principled solution to the weight symmetry problem: our proposed information theoretic criterion aims to maximize the linear dependence between the signals in two neighboring layers, naturally leading to the use of linear or affine transformations in between them. A key property of this approach is that employing two alternative expressions for the correlative mutual information (CMI) results in potentially *asymmetric forward and backward prediction networks*, offering a natural solution to the weight transport problem. Consequently, predictive coding in both directions emerges as the inherent solution to the correlative information maximization principle, fostering signal transmission in both forward and top-down directions through asymmetrical connections. While the CorInfoMax principle enhances information flow in both directions, the introduction of set membership constraints on the layer activations, such as non-negativity, through activation nonlinearities and lateral inhibitions, encourages compression of information and sparse representations [11].
- a normative approach for deriving networks with multi-compartment neurons: the gradient-based optimization of the CorInfoMax objective naturally leads to network models that employ multi-compartment pyramidal neuron models accompanied by interneurons as illustrated in Figure 1.

As derived and explained in detail in Section 2, the resulting networks incorporate lateral connections and auto-synapses (autapses) to increase the entropy of a layer, promoting utilization of all dimensions within the representation space of that layer. Meanwhile, asymmetric feedforward and feedback connections act as forward and backward predictors of layer activation signals, respectively, to reduce the conditional entropies between layers, targeting the elimination of redundancy.

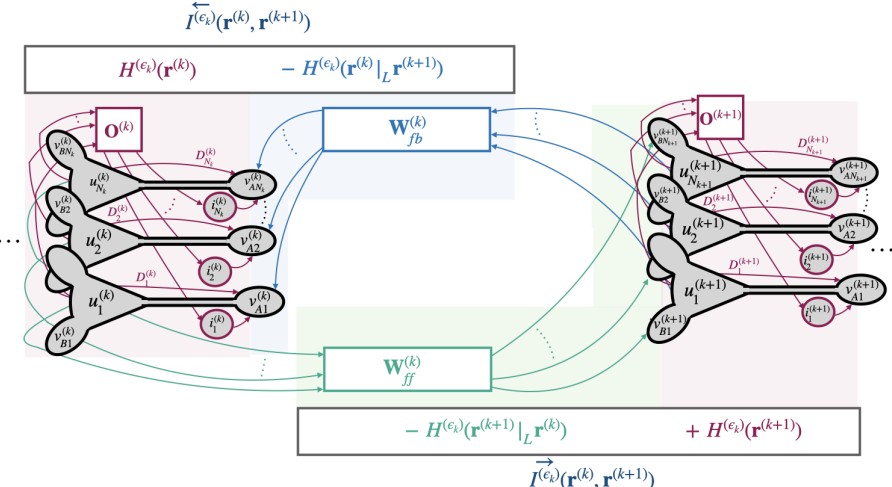

Figure 1: A segment of a correlative information maximization based neural network. Each layer consists of three-compartment pyramidal neurons with outputs $\mathbf{r}^{(k)}$ and membrane voltages ($\mathbf{u}^{(k)}$-soma, $\mathbf{v}_B^{(k)}$-basal dendrites, $\mathbf{v}_A^{(k)}$-distal apical dendrites) and interneurons with outputs $\mathbf{i}^{(k)}$. The CMI expression $\overrightarrow{I^{(\epsilon_k)}}(\mathbf{r}^{(k)}, \mathbf{r}^{(k+1)})$ ($\overleftarrow{I^{(\epsilon_k)}}(\mathbf{r}^{(k)}, \mathbf{r}^{(k+1)})$) defines forward (backward) prediction synapses $\mathbf{W}_{ff}^{(k)}$ ($\mathbf{W}_{fb}^{(k)}$), for minimizing $H^{(\epsilon_k)}(\mathbf{r}^{(k+1)}|_L\mathbf{r}^{(k)})$ ($H^{(\epsilon_k)}(\mathbf{r}^{(k)}|_L\mathbf{r}^{(k+1)})$) and the lateral connections $\mathbf{O}^{(k+1)}$ ($\mathbf{O}^{(k)}$) and autapses $\mathbf{D}^{(k+1)}$ ($\mathbf{D}^{(k)}$) connected to distal apical dendrites at the $k+1$-th ($k$-th) layer, for maximizing $H^{(\epsilon_k)}(\mathbf{r}^{(k+1)})$ ($H^{(\epsilon_k)}(\mathbf{r}^{(k)})$).

## 1.1 Related work

### 1.1.1 Multi-compartmental neuron model based biologically plausible approaches

Experimentally grounded studies, such as [7, 12], have been influential for considering a role for dendritic-processing in multi-compartmental neurons for learning and credit assignment [13]. Subsequent research has explored biologically plausible models with supervised learning functionality, such as the two-compartment neuron model by Urbanczik and Senn [8] and the three-compartment pyramidal neuron model by Sacramento et al. [9]. Both models integrate non-Hebbian learning and spike-time dependent plasticity, while the latter includes SST interneurons [14]. Similar frameworks have been proposed by [15] and [10], with the latter introducing a normative framework based on multi-compartmental neuron structure, top-down feedback, lateral and feedforward connections, and Hebbian and non-Hebbian learning rules, emerging from the optimization of a prediction error objective with a whitening constraint on co-layer neurons.

In a similar vein to [10], we propose an alternative normative framework based on information maximization principle. In this framework, the three-compartment structure and associated forward, top-down and lateral synaptic connections stem from the maximization of CMI between adjacent layers, without the imposition of any whitening constraint.

### 1.1.2 Weight symmetry problem

A central concern regarding the biological plausibility of the backpropagation algorithm pertains to the weight symmetry issue: synaptic weights in the feedback path for error backpropagation are transposes of those used in the forward inference path [2, 3, 16]. The requirement of tied weights in backpropagation is questionable for physically distinct feedforward and feedback paths in biological systems, leading many researchers to focus on addressing the weight symmetry issue.

Various strategies have been devised to address the weight symmetry issue. For example, the feedback alignment approach, which fixes randomly initialized feedback weights and adapts feedforward weights, was offered as a plausible solution [17]. Later Akrout et.al. [18] proposed its extension by updating feedback weights towards to the transpose of the feedforward weights. Along the similar lines, Amit introduced antisymmetry through separate random initializations [19]. Liao et al. [20] showed that the sign of the feedback weights (rather than their magnitude) affects the learning performance, and proposed the sign-symmetry algorithm.

Intriguingly, this symmetric weight structure is also observed in biologically plausible frameworks such as predictive coding (PC) [21, 22, 23], equilibrium propagation (EP) [24, 25, 26], and similarity matching [27]. This phenomenon can be rationalized by the transpose symmetry of the Hebbian update with respect to inputs and outputs. The EP framework in [25] unties forward and backward connections inspired by [28, 29], and only yields small performance degradation. A more recent approach by Golkar et al. [10] addresses this challenge by integrating two alternative forward prediction error loss function terms associated with the same network layer and leveraging presumed whitening constraints to eliminate shared feedback coefficients.

In existing predictive coding-based schemes such as [21, 22, 23], the loss function contains only forward prediction error terms. The feedback connection with symmetric weights, which backpropagates forward prediction error, emerges due to the gradient-based optimization of the PC loss. In contrast, our framework's crucial contribution is the adoption of two alternative expressions for the correlative mutual information between consecutive network layers as the central normative approach. Utilizing these two alternatives naturally leads to both forward and backward prediction paths with asymmetric weights, promoting information flow in both feedforward and top-down directions. Unlike the work of [10], our method circumvents the need for layer whitening constraints and additional forward prediction terms to achieve asymmetric weights.

### 1.1.3 Correlative information maximization

Information maximization has been proposed as a governing or guiding principle in several machine learning and neuroscience frameworks for different tasks: (i) The propagation of information within a self-organized network as pioneered by Linsker [30]. (ii) Extracting hidden features or factors associated with observations by maximizing information between the input and its internal representation such as independent component analysis (ICA-InfoMax) approach by [31]. In the neuroscience

domain, the motivation has been to provide normative explanations to the behaviour of cortical activities evidenced by experimental work, such as orientation and visual stimuli length selectivity of primary visual cortex neurons [32, 33]. The same idea has been recently extended in the machine learning field by the Deep Infomax approach where the goal is to transfer maximum information from the input of a deep network to its final layer, while satisfying prior distribution constraints on the output representations [34]. (iii) Matching representations corresponding to two alternative augmentations or modalities of the same input in the context of self-supervised learning [35].

Correlative mutual information maximization has been recently proposed as an alternative for Shannon Mutual Information (SMI), due to its desirable properties [36]: (i) maximization of CMI is equivalent to maximizing linear dependence, which may be more relevant than establishing arbitrary nonlinear dependence in certain applications [37], (ii) it is based only on the second order statistics, making it relatively easier to optimize. We additionally note that criteria based on correlation are intrinsically linked to local learning rules, leading to biologically plausible implementations, [38, 39]. Erdogan [36] proposed the use of CorInfoMax for solving blind source separation (BSS) problem to retrieve potentially correlated components from their mixtures. Ozsoy et al. [37] proposed maximizing the CMI between the representations of two different augmentations of the same input as a self-supervised learning approach. More recently, Bozkurt et al. [11] introduced an unsupervised framework to generate biologically plausible neural networks for the BSS problem with infinitely many domain selections using the CMI objective.

In this article, we suggest employing the CorInfoMax principle for biologically plausible supervised learning. The key difference compared to the unsupervised framework presented in [11] is the utilization of two alternative forms of mutual information. This leads to a bidirectional information flow that enables error backpropagation without encountering the weight symmetry issue.

## 2 Deep correlative information maximization

### 2.1 Network data model

We assume a dataset with $L$ input data points $\mathbf{x}[t] \in \mathbb{R}^m, t = 1, \ldots, L$, and let $\mathbf{y}_T[t] \in \mathbb{R}^n$ be the corresponding labels. We consider a network with $P-1$ hidden layers whose activities are denoted by $\mathbf{r}^{(k)} \in \mathbb{R}^{N_k}, k = 1, \ldots, P-1$. For notational simplicity, we also denote input and output of the network by $\mathbf{r}^{(0)}$ and $\mathbf{r}^{(P)}$, i.e., $\mathbf{r}^{(0)}[t] = \mathbf{x}[t]$ and $\mathbf{r}^{(P)}[t] = \hat{\mathbf{y}}[t]$. We consider polytopic constraints for the hidden and output layer activities, i.e., $\mathbf{r}^{(k)} \in \mathcal{P}^{(k)}$, where $\mathcal{P}^{(k)}$ is the presumed polytopic domain for the $k$-th layer [11, 40]. We note that the polytopic assumptions are plausible as the activations of neurons in practice are bounded. In particular, we will make the specific assumption that $\mathcal{P}^{(k)} = \mathcal{B}_{\infty,+} = \{\mathbf{r} : \mathbf{0} \preccurlyeq \mathbf{r} \preccurlyeq \mathbf{1}\}$, i.e., (normalized) activations lie in a nonnegative unit-hypercube. Such nonnegativity constraints have been connected to disentangling behavior [41, 42, 43], however, we consider extensions in the form of alternative polytopic sets corresponding to different feature priors [11] (see Appendix C). More broadly, the corresponding label $\mathbf{y}_T$ can be, one-hot encoded label vectors for a classification problem, or discrete or continuous valued vectors for a regression problem.

### 2.2 Correlative information maximization based signal propagation

Our proposed CorInfoMax framework represents a principled approach where both the structure of the network and its internal dynamics as well as the learning rules governing adaptation of its parameters are not predetermined. Instead, these elements emerge naturally from an explicit optimization process. As the optimization objective, we propose the maximization of *correlative mutual information* (see Appendix A between two consecutive network layers. As derived in future sections, the proposed objective facilitates information flow—input-to-output and vice versa, while the presumed domains for the hidden and output layers inherently induce information compression and feature shaping.

In Sections 2.2.1 and 2.2.2, we outline the correlative mutual information-based objective and its implementation based on samples, respectively. Section 2.3 demonstrates that the optimization of this objective through gradient ascent naturally results in recurrent neural networks with multi-compartment neurons. Finally, Section 2.4 explains how the optimization of the same criterion leads to biologically plausible learning dynamics for the resulting network structure.

### 2.2.1 Stochastic CorInfoMax based supervised criterion

We propose the total correlative mutual information among consecutive layers, augmented with the mean-square-error (MSE) training loss, as the stochastic objective to be maximized:

$$J(\mathbf{r}^{(1)}, \ldots, \mathbf{r}^{(P)}) = \sum_{k=0}^{P-1} I^{(\epsilon_k)}(\mathbf{r}^{(k)}, \mathbf{r}^{(k+1)}) - \frac{\beta}{2} E(\|\mathbf{y}_T - \mathbf{r}^{(P)}\|_2^2), \tag{1}$$

where, as defined in [36, 37] and in Appendix A,

$$I^{\overrightarrow{(\epsilon_k)}}(\mathbf{r}^{(k)}, \mathbf{r}^{(k+1)}) = \frac{1}{2} \log \det (\mathbf{R}_{\mathbf{r}^{(k+1)}} + \epsilon_k \boldsymbol{I}) - \frac{1}{2} \log \det \left( \boldsymbol{R}_{\overrightarrow{\mathbf{e}}_*^{(k+1)}} + \epsilon_k \boldsymbol{I} \right), \tag{2}$$

is the correlative mutual information between layers $\mathbf{r}^{(k)}$ an $\mathbf{r}^{(k+1)}$, $\mathbf{R}_{\mathbf{r}^{(k+1)}} = E(\mathbf{r}^{(k+1)}\mathbf{r}^{(k+1)^T})$ is the autocorrelation matrix corresponding to the layer $\mathbf{r}^{(k+1)}$ activations, and $\mathbf{R}_{\overrightarrow{\mathbf{e}}_*^{(k+1)}}$ corresponds to the error autocorrelation matrix for the best linear regularized minimum MSE predictor of $\mathbf{r}^{(k+1)}$ from $\mathbf{r}^{(k)}$. Therefore, the mutual information objective in (2) makes a referral to the *regularized* **forward** *prediction problem* represented by the optimization

$$\underset{\boldsymbol{W}_{ff}^{(k)}}{\text{minimize}} \; E(\|\overrightarrow{\mathbf{e}}^{(k+1)}\|_2^2) + \epsilon_k \|\boldsymbol{W}_{ff}^{(k)}\|_F^2 \quad \text{s.t.} \quad \overrightarrow{\mathbf{e}}^{(k+1)} = \mathbf{r}^{(k+1)} - \boldsymbol{W}_{ff}^{(k)} \mathbf{r}^{(k)}, \tag{3}$$

and $\mathbf{e}_*^{(k+1)}$ is the forward prediction error corresponding to the optimal forward predictor $\boldsymbol{W}_{ff,*}^{(k)}$.

If we interpret the maximization of CMI in (2): the first term on the right side of (2) encourages the spread of $\mathbf{r}^{(k+1)}$ in its presumed domain $\mathcal{P}^{(k+1)}$, while the second term incites the minimization of redundancy in $\mathbf{r}^{(k+1)}$ beyond its component predictable from $\mathbf{r}^{(k)}$.

An equal and alternative expression for the CMI can be written as (Appendix A)

$$I^{\overleftarrow{(\epsilon_k)}}(\mathbf{r}^{(k)}, \mathbf{r}^{(k+1)}) = \frac{1}{2} \log \det(\mathbf{R}_{\mathbf{r}^{(k)}} + \epsilon_k \boldsymbol{I}) - \frac{1}{2} \log \det \left( \boldsymbol{R}_{\overleftarrow{\mathbf{e}}_*^{(k)}} + \epsilon_k \boldsymbol{I} \right), \tag{4}$$

where $\boldsymbol{R}_{\overleftarrow{\mathbf{e}}_*^{(k)}}$ corresponds to the error autocorrelation matrix for the best linear regularized minimum MSE predictor of $\mathbf{r}^{(k)}$ from $\mathbf{r}^{(k+1)}$. The corresponding *regularized* **backward** *prediction problem* is defined by the optimization

$$\underset{\boldsymbol{W}_{fb}^{(k)}}{\text{minimize}} \; E(\|\overleftarrow{\mathbf{e}}^{(k)}\|_2^2) + \epsilon_k \|\boldsymbol{W}_{fb}^{(k)}\|_F^2 \quad \text{s.t.} \quad \overleftarrow{\mathbf{e}}^{(k)} = \mathbf{r}^{(k)} - \boldsymbol{W}_{fb}^{(k)} \mathbf{r}^{(k+1)}. \tag{5}$$

We observe that the two alternative yet equivalent representations of the correlative mutual information between layers $\mathbf{r}^{(k)}$ and $\mathbf{r}^{(k+1)}$ in (2) and (4) are intrinsically linked to the forward and backward prediction problems between these layers, which are represented by the optimizations in (3) and (5), respectively. As we will demonstrate later, the existence of these two alternative forms for the CMI plays a crucial role in deriving a neural network architecture that overcomes the weight symmetry issue.

### 2.2.2 Sample-based supervised CorInfoMax criterion

Our aim is to construct a biologically plausible neural network that optimizes the total CMI, equation (1), in an adaptive manner. Here, we obtain a sample-based version of (1) as a step towards that goal.

We first define the exponentially-weighted sample auto and cross-correlation matrices as follows:

$$\hat{\mathbf{R}}_{\mathbf{r}^{(k)}}[t] = \frac{1 - \lambda_{\mathbf{r}}}{1 - \lambda_{\mathbf{r}}^t} \sum_{i=1}^{t} \lambda_{\mathbf{r}}^{t-i} \mathbf{r}^{(k)}[i]\mathbf{r}^{(k)}[i]^T, \; \hat{\mathbf{R}}_{\mathbf{r}^{(k)}\mathbf{r}^{(k+1)}}[t] = \frac{1 - \lambda_{\mathbf{r}}}{1 - \lambda_{\mathbf{r}}^t} \sum_{i=1}^{t} \lambda_{\mathbf{r}}^{t-i} \mathbf{r}^{(k)}[i]\mathbf{r}^{(k+1)}[i]^T, \tag{6}$$

for $k = 0, \ldots, P$, respectively, where $0 \ll \lambda_{\mathbf{r}} < 1$ is the forgetting factor. Next, we define two equivalent forms of the sample-based CMI, $\hat{I}^{(\epsilon)}(\mathbf{r}^{(k)}, \mathbf{r}^{(k+1)})[t]$:

$$\hat{I}^{\overrightarrow{(\epsilon_k)}}(\mathbf{r}^{(k)}, \mathbf{r}^{(k+1)})[t] = \frac{1}{2} \log \det(\hat{\mathbf{R}}_{\mathbf{r}^{(k+1)}}[t] + \epsilon_k \boldsymbol{I}) - \frac{1}{2} \log \det(\hat{\mathbf{R}}_{\overrightarrow{\mathbf{e}}_*^{(k+1)}}[t] + \epsilon_k \boldsymbol{I}), \tag{7}$$

$$\hat{I}^{\overleftarrow{(\epsilon_k)}}(\mathbf{r}^{(k)}, \mathbf{r}^{(k+1)})[t] = \frac{1}{2} \log \det(\hat{\mathbf{R}}_{\mathbf{r}^{(k)}}[t] + \epsilon_k \boldsymbol{I}) - \frac{1}{2} \log \det(\hat{\boldsymbol{R}}_{\overleftarrow{\mathbf{e}}_*^{(k)}}[t] + \epsilon_k \boldsymbol{I}), \tag{8}$$

where $\hat{\boldsymbol{R}}_{\overrightarrow{\mathbf{e}}_*^{(k+1)}}[t]$ is the exponentially-weighted sample autocorrelation matrix for the forward prediction error at level-$(k+1)$, $\overrightarrow{\mathbf{e}}^{(k+1)*}[t]$, corresponding to the best linear exponentially-weighted regularized least squares predictor of $\mathbf{r}^{(k+1)}[t]$ from the lower level activations $\mathbf{r}^{(k)}[t]$. Similarly, $\hat{\boldsymbol{R}}_{\overleftarrow{\mathbf{e}}^{(k)}}[t]$ is the exponentially-weighted autocorrelation matrix for the backward prediction error at level-$(k)$, $\overleftarrow{\mathbf{e}}^{(k)}[t]$, corresponding to the best linear exponentially-weighted regularized least squares predictor of $\mathbf{r}^{(k)}[t]$ from the higher level activations $\mathbf{r}^{(k+1)}[t]$.

The sample-based CorInfoMax optimization can be written as:

$$\underset{\mathbf{r}^{(k)}[t],\, k=0,\ldots,P}{\text{maximize}} \quad \sum_{k=0}^{P-1} \hat{I}^{(\epsilon_k)}(\mathbf{r}^{(k)}, \mathbf{r}^{(k+1)})[t] - \frac{\beta}{2}\|\mathbf{y}_T[t] - \mathbf{r}^{(P)}[t]\|_2^2 = \hat{J}(\mathbf{r}^{(1)}, \ldots, \mathbf{r}^{(P)})[t] \tag{9a}$$

$$\text{subject to} \quad \mathbf{r}^{(k)}[t] \in \mathcal{P}^{(k)}, k = 1, \ldots, P, \tag{9b}$$

$$\mathbf{r}^{(0)}[t] = \mathbf{x}[t], \tag{9c}$$

As outlined in Appendix B, we can employ Taylor series linearization to approximate the $\log\det$ terms associated with forward and backward prediction errors in (2) and (4) in the form

$$\log\det\left(\hat{\boldsymbol{R}}_{\overrightarrow{\mathbf{e}}^{(k+1)}}[t] + \epsilon_k \boldsymbol{I}\right)$$

$$\approx \frac{1}{\epsilon_k}\sum_{i=1}^{t} \lambda_{\mathbf{r}}^{t-i}\|\mathbf{r}^{(k+1)}[i] - \boldsymbol{W}_{ff,*}^{(k)}[t]\mathbf{r}^{(k)}[i]\|_2^2 + \epsilon_k\|\boldsymbol{W}_{ff,*}^{(k)}[t]\|_F^2 + N_{k+1}\log(\epsilon_k) \tag{10}$$

$$\log\det\left(\hat{\boldsymbol{R}}_{\overleftarrow{\mathbf{e}}^{(k)}}[t] + \epsilon_k \boldsymbol{I}\right)$$

$$\approx \frac{1}{\epsilon_k}\sum_{i=1}^{t} \lambda_{\mathbf{r}}^{t-i}\|\mathbf{r}^{(k)}[i] - \boldsymbol{W}_{fb,*}^{(k)}[t]\mathbf{r}^{(k+1)}[i]\|_2^2 + \epsilon_k\|\boldsymbol{W}_{fb,*}^{(k)}[t]\|_F^2 + N_k\log(\epsilon_k), \tag{11}$$

where $\mathbf{W}_{ff,*}^{(k)}[t]$ is the optimal linear regularized weighted least squares forward predictor coefficients in predicting $\mathbf{r}^{(k+1)}[i]$ from $\mathbf{r}^{(k)}[i]$ for $i = 1, \ldots, t$, and $\mathbf{W}_{fb,*}^{(k)}[t]$ is the optimal linear regularized weighted least squares backward predictor coefficients in predicting $\mathbf{r}^{(k)}[i]$ from $\mathbf{r}^{(k+1)}[i]$ for $i = 1, \ldots, t$. Consequently, the optimal choices of forward and backward predictor coefficients are coupled with the optimal choices of layer activations.

In the online optimization process, we initially relax the requirement on the optimality of predictors and start with random predictor coefficient selections. During the learning process, we apply a coordinate ascent-based procedure on activation signals and predictor coefficients. Specifically, at time step-$t$, we consider two phases:

1. First, we optimize with respect to the activations $\{\mathbf{r}^{(k)}[t], k = 1, \ldots, P\}$, where we assume predictor coefficients to be fixed. This phase yields network structure and output dynamics,

2. Next, we update the forward and backward predictor coefficients $\boldsymbol{W}_{ff}^{(k)}$ and $\boldsymbol{W}_{fb}^{(k)}$, for $k = 1, \ldots, P$, to reduce the corresponding forward and backward prediction errors, respectively. This phase provides update expressions to be utilized in learning dynamics.

As the algorithm iterations progress, the predictor coefficients converge to the vicinity of their optimal values.

For the first phase of the online optimization, we employ a projected gradient ascent-based approach for activations: for $k = 1, \ldots, P-1$, the layer activation vector $\mathbf{r}^{(k)}[t]$ is included in the objective function terms $\hat{I}^{(\epsilon)}(\mathbf{r}^{(k-1)}, \mathbf{r}^{(k)})[t]$ and $\hat{I}^{(\epsilon)}(\mathbf{r}^{(k)}, \mathbf{r}^{(k+1)})[t]$. Therefore, to calculate the gradient with respect to $\mathbf{r}^{(k)}[t]$, we can use expressions in (7) and (8). More specifically, we choose $\hat{J}_k(\mathbf{r}^{(k)})[t] = \overrightarrow{\hat{I}^{(\epsilon_{k-1})}}(\mathbf{r}^{(k-1)}, \mathbf{r}^{(k)})[t] + \overleftarrow{\hat{I}^{(\epsilon_k)}}(\mathbf{r}^{(k)}, \mathbf{r}^{(k+1)})[t]$ for $k = 1, \ldots, P-1$, to represent the components of the objective function in (9a) involving $\mathbf{r}^{(k)}[t]$. As described in Appendix G.1, this choice is instrumental in avoiding weight transport problem. Similarly, we can write the component of

the objective function in (9a) that is dependent on the final layer activations as $\hat{J}_P(\mathbf{r}^{(P)})[t] = \hat{I}^{(\epsilon_{P-1})}(\mathbf{r}^{(P-1)}, \mathbf{r}^{(P)})[t] - \frac{\beta}{2}\|\mathbf{r}^{(P)}[t] - \mathbf{y}_T[t]\|_2^2$.

Based on the derivations presented in Appendix D, which directly incorporate the approximations from (10) and (11), we can express the gradient of the objective function in (9a) with respect to $\mathbf{r}^{(k)}$, for $k = 1, \ldots, P - 1$ as:

$$\nabla_{\mathbf{r}^{(k)}}\hat{J}(\mathbf{r}^{(1)}, \ldots, \mathbf{r}^{(P)})[t] = 2\gamma \boldsymbol{B}_{\mathbf{r}^{(k)}}[t]\mathbf{r}^{(k)}[t] - \frac{1}{\epsilon_{k-1}}\overrightarrow{\mathbf{e}}^{(k)}[t] - \frac{1}{\epsilon_k}\overleftarrow{\mathbf{e}}^{(k)}[t], \qquad (12)$$

where $\gamma = \frac{1-\lambda_{\mathbf{r}}}{\lambda_{\mathbf{r}}}$,

$$\overrightarrow{\mathbf{e}}^{(k)}[t] = \mathbf{r}^{(k)}[t] - \boldsymbol{W}_{ff}^{(k-1)}[t]\mathbf{r}^{(k-1)}[t], \quad \overleftarrow{\mathbf{e}}^{(k)}[t] = \mathbf{r}^{(k)}[t] - \boldsymbol{W}_{fb}^{(k)}[t]\mathbf{r}^{(k+1)}[t], \qquad (13)$$

and $\boldsymbol{B}_{\mathbf{r}^{(k)}}[t] = (\hat{\mathbf{R}}_{\mathbf{r}^{(k)}}[t] + \epsilon_{k-1}\boldsymbol{I})^{-1} \approx (\hat{\mathbf{R}}_{\mathbf{r}^{(k)}}[t] + \epsilon_k\boldsymbol{I})^{-1}$. Similarly, for $k = P$, we have

$$\nabla_{\mathbf{r}^{(P)}}\hat{J}(\mathbf{r}^{(1)}, \ldots, \mathbf{r}^{(P)})[t] = \gamma \boldsymbol{B}_{\mathbf{r}^{(P)}}[t]\mathbf{r}^{(P)}[t] - \frac{1}{\epsilon_{P-1}}\overrightarrow{\mathbf{e}}^{(P)}[t] - \beta(\mathbf{r}^{(P)}[t] - \mathbf{y}_T[t]). \qquad (14)$$

## 2.3 Neural network formulation based on information maximization

In this section, we develop a biologically plausible neural network grounded on the correlative information maximization-based network propagation model outlined in Section 2.2. To achieve this, we employ projected gradient ascent optimization for determining layer activations $\mathbf{r}^{(1)}[t], \mathbf{r}^{(2)}[t], \ldots, \mathbf{r}^{(P)}[t]$, which shape the network structure and dynamics, as well as updating the corresponding synapses that govern the learning dynamics.

### 2.3.1 Network structure and neural dynamics

In this section, we show that the projected gradient ascent solution to the optimization in (9) defines a multilayer recurrent neural network. To this end, we introduce the intermediate variable $\mathbf{u}^{(k)}$ as the updated layer-$k$ activations prior to the projection onto the domain set $\mathcal{P}^{(k)}$. Utilizing the gradient expressions in (12)-(13), we can express the network dynamics for layers $k = 1, \ldots, P - 1$ as follows (see Appendix E for details):

$$\tau_{\mathbf{u}}\frac{d\mathbf{u}^{(k)}[t; s]}{ds} = -g_{lk}\mathbf{u}^{(k)}[t; s] + \frac{1}{\epsilon_k}\boldsymbol{M}^{(k)}[t]\boldsymbol{r}^{(k)}[t; s] - \frac{1}{\epsilon_{k-1}}\overrightarrow{\mathbf{e}}_u^{(k)}[t; s] - \frac{1}{\epsilon_k}\overleftarrow{\mathbf{e}}_u^{(k)}[t; s], \qquad (15)$$

$$\overrightarrow{\mathbf{e}}_u^{(k)}[t; s] = \mathbf{u}^{(k)}[t; s] - \boldsymbol{W}_{ff}^{(k-1)}[t]\mathbf{r}^{(k-1)}[t; s], \quad \overleftarrow{\mathbf{e}}_u^{(k)}[t; s] = \mathbf{u}^{(k)}[t; s] - \boldsymbol{W}_{fb}^{(k)}[t]\mathbf{r}^{(k+1)}[t; s], \qquad (16)$$

$$\mathbf{r}^{(k)}[t; s] = \sigma_+(\mathbf{u}^{(k)}[t; s]), \qquad (17)$$

where $t$ is the discrete data index, $s$ is the continuous time index corresponding to network dynamics, $\tau_{\mathbf{u}}$ is the update time constant, $\boldsymbol{M}^{(k)}[t] = \epsilon_k(2\gamma\boldsymbol{B}_{\mathbf{r}^{(k)}}[t] + g_{lk}\boldsymbol{I})$, and $\sigma_+$ represents the elementwise clipped-ReLU function corresponding to the projection onto the nonnegative unit-hypercube $\mathcal{B}_{\infty,+}$, defined as $\sigma_+(u) = \min(1, \max(u, 0))$.

To reinterpret the dynamics in (15) to (17) as a multi-compartmental neural network, for $k = 1, \ldots, P - 1$, we define the signals:

$$\mathbf{v}_A^{(k)}[t; s] = \boldsymbol{M}^{(k)}[t]\boldsymbol{r}^{(k)}[t; s] + \boldsymbol{W}_{fb}^{(k)}[t]\mathbf{r}^{(k+1)}[t; s], \quad \mathbf{v}_B^{(k)}[t; s] = \boldsymbol{W}_{ff}^{(k-1)}[t]\mathbf{r}^{(k-1)}[t; s], \qquad (18)$$

which allow us to rewrite the network activation dynamics (15) to (17) as:

$$\tau_{\mathbf{u}}\frac{d\mathbf{u}^{(k)}[t; s]}{ds} = -g_{lk}\mathbf{u}^{(k)}[t; s] + g_{A,k}(\mathbf{v}_A^{(k)}[t; s] - \mathbf{u}^{(k)}[t; s]) + g_{B,k}(\mathbf{v}_B^{(k)}[t; s] - \mathbf{u}^{(k)}[t; s]), \quad (19)$$

$$\mathbf{r}^{(k)}[t; s] = \sigma_+(\mathbf{u}^{(k)}[t; s]), \qquad (20)$$

where $g_{A,k} = \frac{1}{\epsilon_{k-1}}$ and $g_{B,k} = \frac{1}{\epsilon_k}$. Similarly, for the output layer, we employ the same expressions as (19) and (20) with $k = P$, except that in this case we have:

$$\mathbf{v}_A^{(P)}[t; s] = \boldsymbol{M}^{(P)}[t]\boldsymbol{r}^{(k)}[t; s] - (\mathbf{r}^{(P)}[t; s] - \mathbf{y}_T[t]), \quad \mathbf{v}_B^{(P)}[t; s] = \boldsymbol{W}_{ff}^{(P-1)}[t]\mathbf{r}^{(P-1)}[t; s], \quad (21)$$

where $g_{B,P} = \frac{1}{\epsilon_{P-1}}$, $g_{A,P} = \beta$ and $\boldsymbol{M}^{(P)}[t] = \beta^{-1}(\gamma \boldsymbol{B}_{\mathbf{r}^{(P)}}[t] + g_{lk}\boldsymbol{I})$.

Remarkably, the equations (18) to (21) reveal a biologically plausible neural network that incorporates three-compartment pyramidal neuron models, as presented in [9, 10]. This intricate architecture, of which two-layer segment is demonstrated in Figure 1, naturally emerges from the proposed correlative information maximization framework. In this network structure:

- $\mathbf{u}^{(k)}$ embodies the membrane potentials for neuronal somatic compartments of the neurons at layer-$k$, where $\tau_{\mathbf{u}}$ is the membrane leak time constant of soma.

- $\mathbf{v}_B^{(k)}$ corresponds to membrane potentials for basal dendrite compartments, receiving feedforward input originating from the previous layer.

- $\mathbf{v}_A^{(k)}$ denotes the membrane potentials for distal apical dendrite compartments, which gather top-down input from the subsequent layer and lateral inputs represented by $\boldsymbol{M}^{(k)}[t]\boldsymbol{r}^{(k)}$ in (18) and (21). Decomposing $\boldsymbol{M}^{(k)}$ into $\boldsymbol{D}^{(k)} - \boldsymbol{O}^{(k)}$, we find that $\boldsymbol{D}^{(k)}$ mirrors autapses[44], and the off-diagonal component $\boldsymbol{O}^{(k)}$ corresponds to lateral inhibition synapses. We use $\mathbf{i}^{(k)} = -\boldsymbol{O}^{(k)}\mathbf{r}^{(k)}$ to represent the activations of SST interneurons [14] that generate lateral inhibitions to the apical dendrites.

- Forward (backward) prediction errors manifest in the membrane voltage differences between soma and basal (distal) compartments of the pyramidal neurons.

- Forward (backward) prediction coefficients $\boldsymbol{W}_{ff}^{(k)}$ ($\boldsymbol{W}_{fb}^{(k)}$) are associated with feedforward (top-down) synapses connecting layers $(k)$ and $(k+1)$.

- The inverse of the regularization coefficient $\epsilon_k$ is related to the conductance between soma and dendritic compartments. This is compliant with the interpretation of the $\epsilon^{-1}$ in Appendix A.2 as the sensitivity parameter that determines the contribution of the prediction errors to the CMI. Conversely, at the output layer, the augmentation constant $\beta$ corresponds to the conductance between soma and distal compartments. This relationship can be motivated by modifying the objective in (9a) as

$$\sum_{k=0}^{P-1} \hat{I}^{(\epsilon_k)}(\mathbf{r}^{(k)}, \mathbf{r}^{(k+1)})[t] + \frac{1}{2}\hat{I}^{\overleftarrow{(\beta^{-1})}}(\mathbf{r}^{(P)}, \mathbf{y}_T)[t], \tag{22}$$

where, through the first-order approximation, the $\mathbf{r}^{(P)}[t]$ dependent portion of $\hat{I}^{\overleftarrow{(\beta^{-1})}}(\mathbf{r}^{(P)}, \mathbf{y}_T)[t]$ can be expressed as $-\beta\|\mathbf{r}^{(P)}[t] - \boldsymbol{W}_{fb}^{(P)}\mathbf{y}_T[t]\|_2^2$. For accuracy, we enforce $\boldsymbol{W}_{fb}^{(P)} = \boldsymbol{I}$.

## 2.4 Learning dynamics

Network parameters consists of feedforward $\boldsymbol{W}_{ff}^{(k)}$, feedback $\boldsymbol{W}_{fb}^{(k)}$ and lateral $\boldsymbol{B}^{(k)}$ coefficients. The learning dynamics of these coefficients are elaborated below:

- *Feedforward Coefficients* are connected to the forward prediction problem defined by the optimization in (3). We can define the corresponding online optimization objective function as $C_{ff}(\boldsymbol{W}_{ff}^{(k)}) = \epsilon_k\|\boldsymbol{W}_{ff}^{(k)}\|_F^2 + \|\overrightarrow{\mathbf{e}}^{(k+1)}[t]\|_2^2$ for which the the partial derivative is given by

$$\frac{\partial C_{ff}(\boldsymbol{W}_{ff}^{(k)}[t])}{\partial \boldsymbol{W}_{ff}^{(k)}} = 2\epsilon_k\boldsymbol{W}_{ff}^{(k)}[t] - 2\overrightarrow{\mathbf{e}}^{(k+1)}[t]\mathbf{r}^{(k)}[t]^T. \tag{23}$$

In Appendix H, we provide a discussion on rewriting (23) in terms of the membrane voltage difference between the distal apical and soma compartments of the neuron, based on the equilibrium condition for the neuronal dynamics:

$$-\overrightarrow{\mathbf{e}}^{(k+1)}[t]\mathbf{r}^{(k)}[t]^T = g_{B,k}^{-1}(g_{A,k}\mathbf{v}_A^{(k)}[t] - (g_{lk} + g_{A_k})\mathbf{u}_*^{(k)}[t] + \mathbf{h}_*[t])\mathbf{r}^{(k)}[t]^T, \tag{24}$$

where $\mathbf{h}_*[t]$ is nonzero only for neurons that are silent or firing at the maximum rate.

- Similarly, *Feedback Coefficients* are connected to the backward prediction problem defined by the optimization in (5), and the corresponding online optimization objective function as $C_{fb}(\boldsymbol{W}_{fb}^{(k)}) = \epsilon_k\|\boldsymbol{W}_{ff}^{(k)}\|_F^2 + \|\overleftarrow{\mathbf{e}}^{(k)}[t]\|_2^2$ for which the partial derivative is given by

$$\frac{\partial C_{fb}(\boldsymbol{W}_{fb}^{(k)}[t])}{\partial \boldsymbol{W}_{fb}^{(k)}} = 2\epsilon_k\boldsymbol{W}_{fb}^{(k)}[t] - 2\overleftarrow{\mathbf{e}}^{(k)}[t]\mathbf{r}^{(k+1)}[t]^T. \tag{25}$$

To compute the updates of both feedforward and feedback coefficients, we use the EP approach [24], where the update terms are obtained based on the contrastive expressions of partial derivatives in (23) and (25) for the nudge phase, i.e., $\beta = \beta' > 0$, and the free phase, i.e., $\beta = 0$, :

$$\delta \boldsymbol{W}_{ff}^{(k)}[t] \propto \frac{1}{\beta'} \left( (\overrightarrow{\mathbf{e}}^{(k+1)}[t]\mathbf{r}^{(k)}[t]^T) \Big|_{\beta=\beta'} - (\overrightarrow{\mathbf{e}}^{(k+1)}[t]\mathbf{r}^{(k)}[t]^T) \Big|_{\beta=0} \right), \tag{26}$$

$$\delta \boldsymbol{W}_{fb}^{(k)}[t] \propto \frac{1}{\beta'} \left( (\overleftarrow{\mathbf{e}}^{(k)}[t]\mathbf{r}^{(k+1)}[t]^T) \Big|_{\beta=\beta'} - (\overleftarrow{\mathbf{e}}^{(k)}[t]\mathbf{r}^{(k+1)}[t]^T) \Big|_{\beta=0} \right). \tag{27}$$

- *Lateral Coefficients*, $\mathbf{B}^{(k)}$ are the inverses of the $\epsilon \mathbf{I}$ perturbed correlation matrices. We can use the update rule in [11] for their learning dynamics after the nudge phase:

$$\boldsymbol{B}^{(k)}[t+1] = \lambda_{\mathbf{r}}^{-1}(\boldsymbol{B}^{(k)}[t] - \gamma \mathbf{z}^{(k)}[t]\mathbf{z}^{(k)}[t]^T), \text{ where } \mathbf{z}^{(k)} = \boldsymbol{B}^{(k)}[t]\mathbf{r}^{(k)}[t]\big|_{\beta=\beta'}. \tag{28}$$

As we derived in Appendix F, we can rewrite the update rule of the lateral weights in terms of the updates of autapses and lateral inhibition synapses as follows:

$$\mathbf{D}_{ii}^{(k)}[t+1] = \lambda_{\mathbf{r}}^{-1}\mathbf{D}_{ii}^{(k)}[t] - \lambda_{\mathbf{r}}^{-1}\epsilon_k 2\gamma^2 (\mathbf{z}_i^{(k)}[t])^2 + \epsilon_k g_{lk}(1 - \lambda_{\mathbf{r}}^{-1}), \quad \forall i \in \{1, \ldots, N_k\} \tag{29}$$

$$\mathbf{O}_{ij}^{(k)}[t+1] = \lambda_{\mathbf{r}}^{-1}\mathbf{O}^{(k)}[t]_{ij} + \lambda_{\mathbf{r}}^{-1}\epsilon_k 2\gamma^2 \mathbf{z}_i^{(k)}[t]\mathbf{z}_j^{(k)}[t], \quad \forall i,j \in \{1, \ldots, N_k\}, \text{ where } i \neq j \tag{30}$$

## 3 Discussion of results

- In (A.14), we devise an update for layer activation $\mathbf{r}^{(k)}$ by employing two distinct forms of the CMI associated with $\mathbf{r}^{(k)}$: $\hat{I}^{(\epsilon_{k-1})}(\mathbf{r}^{(k-1)}, \mathbf{r}^{(k)})[t]$, the CMI with the preceding layer, encompassing the forward prediction error for estimating $\mathbf{r}^{(k)}$, and $\hat{I}^{(\epsilon_k)}(\mathbf{r}^{(k)}, \mathbf{r}^{(k+1)})[t]$, the CMI with the subsequent layer, incorporating the backward prediction error for estimating $\mathbf{r}^{(k)}$. Employing these alternative expressions is crucial in circumventing the weight transport problem and offering a more biologically plausible framework. For further discussion, please refer to Appendix G.

- In the context of the proposed correlative information maximization framework, forward and backward predictive coding naturally emerges as a crucial mechanism. By incorporating both alternative expressions of CMI, the framework focuses on minimizing both forward and backward prediction errors between adjacent layers via feedforward and feedback connections. These connections foster bidirectional information flow, thereby enhancing the overall learning process.

- Figure 1 depicts the interplay between the CorInfoMax objective and the corresponding network architecture. The emergence of lateral connections and autapses can be attributed to the maximization of the unconditional layer entropy component of the CMI, which allows for efficient utilization of the available representation dimensions and avoids dimensional degeneracy. Simultaneously, the minimization of conditional entropies between adjacent layers gives rise to feedforward and feedback connections, effectively reducing redundancy within representations.

- We employ time-contrastive learning, as in GenRec [45], EP [24] and CSM [27], by implementing separate phases with Hebbian and anti-Hebbian updates, governed by an assumed teaching signal. It has been conjectured that the teaching signal in biological networks can be modeled by the oscillations in the brain [2, 46, 47]. Although the oscillatory rhythms and their synchronization in the brain are elusive, they are believed to play an important role in adaptive processes such as learning and predicting upcoming events [48, 49].

## 4 Numerical experiments

In this section, we evaluate the performance of our CorInfoMax framework with two layer fully connected networks on image classification tasks using three popular datasets: MNIST [50], Fashion-MNIST [51], and CIFAR10 [52]. We used layer sizes of $784, 500, 10$ for both MNIST and Fashion-MNIST datasets while we used layer sizes of $3072, 1000, 10$ for CIFAR10 dataset, and the final layer size 10 corresponds to one-hot encoded ouput vectors. Further details including full set of hyperparameters can be found in Appendix J. We compare the effectiveness of our approach against

other contrastive methods, such as EP [24] and CSM [27], as well as explicit methods, including PC [22] and PC-Nudge [53], when training multilayer perceptron (MLP) architectures.

We examine two distinct constraints on the activations of CorInfoMax Networks: (i) $\mathcal{B}_{\infty,+}$, representing the nonnegative part of the unit hypercube, and (ii) $\mathcal{B}_{1,+} = \{\mathbf{r} : \mathbf{r} \succcurlyeq 0, \|\mathbf{r}\|_1 \leq 1\}$, denoting the nonnegative part of the unit $\ell_1$-norm ball [40]. Table 1 presents the test accuracy results for each algorithm, averaged over 10 realizations along with the corresponding standard deviations. These findings demonstrate that CorInfoMax networks can achieve comparable or superior performance in relation to the state-of-the-art methods for the selected tasks. Additional information regarding these experiments, as well as further experiments, can be found in the Appendix. Our code is available online[1].

Table 1: Test accuracy results (mean $\pm$ standard deviation from $n = 10$ runs) for CorInfoMax networks are compared with other biologically-plausible algorithms. The performance of CSM on the CIFAR10 dataset is taken from [27], while the remaining results stem from our own simulations.

| | MNIST | FashionMNIST | CIFAR10 |
|---|---|---|---|
| **CorInfoMax-**$\mathcal{B}_{\infty,+}$ (Appendix J.4) | $97.62 \pm 0.1$ | $88.14 \pm 0.3$ | $51.86 \pm 0.3$ |
| **CorInfoMax-**$\mathcal{B}_{1,+}$ (Appendix J.6) | $97.71 \pm 0.1$ | $88.09 \pm 0.1$ | $51.19 \pm 0.4$ |
| EP | $97.61 \pm 0.1$ | $88.06 \pm 0.7$ | $49.28 \pm 0.5$ |
| CSM | $98.08 \pm 0.1$ | $88.73 \pm 0.2$ | $40.79^*$ |
| PC | $98.17 \pm 0.2$ | $89.31 \pm 0.4$ | - |
| PC-Nudge | $97.71 \pm 0.1$ | $88.49 \pm 0.3$ | $48.58 \pm 0.7$ |
| Feedback Alignment (with MSE Loss) | $97.99 \pm 0.03$ | $88.72 \pm 0.5$ | $50.75 \pm 0.4$ |
| Feedback Alignment (with CrossEntropy Loss) | $97.95 \pm 0.08$ | $88.38 \pm 0.9$ | $52.37 \pm 0.4$ |
| BP (with MSE Loss) | $97.58 \pm 0.01$ | $88.39 \pm 0.1$ | $52.75 \pm 0.1$ |
| BP (with CrossEntropy Loss) | $98.27 \pm 0.03$ | $89.41 \pm 0.2$ | $53.96 \pm 0.3$ |

## 5 Discussion and Conclusion

In this article, we have presented the correlative information maximization (CorInfoMax) framework as a biologically plausible approach to constructing supervised neural network models. Our proposed method addresses the long-standing weight symmetry issue by providing a principled solution, which results in asymmetric forward and backward prediction networks. The experimental analyses demonstrates that CorInfoMax networks provide better or on-par performance in image classification tasks compared to other biologically plausible networks while alleviating the weight symmetry problem. Furthermore, the CorInfoMax framework offers a normative approach for developing network models that incorporate multi-compartment pyramidal neuron models, aligning more closely with the experimental findings about the biological neural networks. The proposed framework is useful in obtaining potential insights such as the role of lateral connections in embedding space expansion and avoiding degeneracy, feedback and feedforward connections for prediction to reduce redundancy, and activation functions/interneurons to shape feature space and compress. Despite the emphasis on supervised deep neural networks in our work, it's crucial to highlight that our approach—replacing the backpropagation algorithm, which suffers from the weight transportation problem, with a normative method devoid of such issues—is potentially extendable to unsupervised and self-supervised learning contexts.

One potential limitation of our framework, shared by other supervised approaches, is the necessity for model parameter search to improve accuracy. We discuss this issue in detail in Appendix K. Another limitation stems from the intrinsic nature of our approach, which involves the determination of neural activities through recursive dynamics (see Appendix J). While this aspect is fundamental to our methodology, it does result in slower computation times compared to conventional neural networks in digital hardware implementation. However, it is worth noting that our proposed network, characterized by local learning rules, holds the potential for efficient and low-power implementations on future neuromorphic hardware chips. Furthermore, our method employs the time contrastive learning technique known as Equilibrium Propagation, which necessitates two distinct phases for learning.

---

[1]https://github.com/BariscanBozkurt/Supervised-CorInfoMax

## 6 Acknowledgments and Disclosure of Funding

This research was supported by KUIS AI Center Research Award. B. Bozkurt acknowledges the support by Gatsby PhD programme, which is supported by the Gatsby Charitable Foundation (GAT3850). C. Pehlevan is supported by NSF Award DMS-2134157, NSF CAREER Award IIS-2239780, and a Sloan Research Fellowship. This work has been made possible in part by a gift from the Chan Zuckerberg Initiative Foundation to establish the Kempner Institute for the Study of Natural and Artificial Intelligence.

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
