# OpenReview forum: "Correlative Information Maximization: A Biologically Plausible Approach to Supervised Deep Neural Networks without Weight Symmetry"
_NeurIPS.cc/2023/Conference — NeurIPS 2023 poster_

### Official Review · Reviewer_wEH2 · 2023-06-30

**Soundness:** 2 fair
**Presentation:** 2 fair
**Contribution:** 2 fair
**Rating:** 4
**Confidence:** 3

**Summary:**

This paper seeks to present a biologically plausible learning approach for supervised learning in deep neural networks. Unlike backpropagation, the approach does not require symmetric weights in the forward and backward directions. The approach relies on an information theoretic approach which seeks to maximize mutual information between layers in the forward and backward direction. The approach is demonstrated on simple data sets (e.g. MNIST) as well as on 3-compartment models of pyramidal neurons.

**Strengths:**

The manuscript addresses an important problem--essentially how can supervised learning be implemented in biological neural networks. It proposes a solution to the well-known weight symmetry/transport problem. And it seeks to do so in a principled way using information theoretic notions

**Weaknesses:**

Unfortunately, the paper is poorly written, with heavy notations and equations which often obscure the approach rather than clarifying it.

There is no clear expression for the learning algorithm. It is hard to see that the learning algorithm is local both in space and time, which is a major requirement in a biologically plausible network.

For the experiments, the authors report the test accuracy. However other metrics would also be interesting, for instance, the degree of symmetry between the final weights in both directions.

It seems that the approach still requires propagating error information over long distances (across many layers) which may also be problematic from a biological point of view.

Supervised learning is not particularly biologically plausible.  This point should be addressed, as a minimum using self-supervised learning in combination with the proposed approach.

The authors should mention more clearly that the weight transport problem is completely solved by random backpropagation or feedback alignment. Thus the advantages of their approach, if any, should be contrasted with feedback alignment.

**Questions:**

It is not clear what you do at the top layer with the targets and the error.  Are you clamping the output layer to the targets and then applying your backward pass?

The term "disputed" in the abstract is too weak. I think there is a strong general consensus that plain backpropagation is not biologically plausible.

line 42: "maximize the linear dependence"--why linear, since the networks are typically non-linear? furthermore it is easy to maximize linear dependence by making the signals identical (i.e. identical activities in two neighboring layers of the same size, which is NOT interesting.

line 59: "predictors"  of what?

Figure 1 is not easy to understand at this stage of the paper. In fact, the use of compartmental models at the beginning of the paper is confusing. Maybe these should be moved entirely toward the end of the paper or in an appendix to improve readability.

To the best of our knowledge, the result by Liao et al is not reliable --it has not been confirmed systematically.

The term polytopic is not very common and should be defined. Is this assumption needed in the case of artificial neural networks?

line 148: why do you mention that the objective is stochastic, and stochastic in which sense? Also, to be clear, you should specify whether you are trying to maximize or minimize the objective.

Is there some Gaussian assumption behind equation 2 (if so it should be stated explicitly). Also, any connection or divergence from better-known concepts, such as mutual information, should be clarified.

Equation 2 and what follows are particularly unclear and hard to follow.

153:  "problem" Which problem exactly?

page 6: the learning rule is not clear. Can you write it in the standard form of  "\delta w_{ij}=learning rate x update urle....$?

**Limitations:**

See some of the remarks above. There is no discussion of the limitations of the proposed approach.

---

> ### Author Rebuttal · Authors · 2023-08-08
>
> We thank you for your detailed review and  useful feedback. We  had to truncate some of our responses to adhere to the length constraints.  We are very eager to provide more details for any questions you might have during the discussion period.
> >..poorly written..heavy notations..
>
> We acknowledge your feedback concerning the accessibility. In the revision, we refined the presentation in Section 2 by relocating certain equations to appendix and providing more comprehensive explanations for enhanced clarity. For a detailed outline of these changes, please refer to items 1 to 3 in our global rebuttal response (GRR).
> >..no clear expression for learning..hard to see the algorithm is local..
>
> In initial submission, we presented the learning rule (see Eq. 26-27) with limited details due to space constraints. To rectify this, we've enriched our revision with an appendix section, "Learning Dynamics", with succinct pseudocode for our algorithm and additional elaboration, ensuring the locality constraints are clearly met. Refer to Alg. 1 in our global rebuttal PDF for more details.
> >..other metrics would be interesting, e.g., the degree of symmetry..
>
> We indeed evaluated the angle between forward and backward weights using the cosine angle metric (Eq. A.7), detailed in our appendix. Refer to Fig. 2 (App. B.3) for the angle's evolution in our 2-layer experiments, starting with $90^\circ$ converging to $\sim70^\circ$. Figs 10&15 illustrate the angle patterns in 3-layer case and sparse networks, highlighting the inherent asymmetry for our framework.
> >..prop. error information over long distances..
>
> The CorInfoMax networks  solely include feedback projections from the next layer in the hierarchy, aligning with known biological connectivity patterns. They lack a separate error prop. mechanism or direct long-range top-down projections
> >..Supervised learning is not particularly bio-plausible..
>
> We recognize the essential role of unsupervised/self-supervised learning in natural learning processes. Our main contribution lies in the dynamics and structure of info. propagation, which can be extended to unsupervised objectives. This point will be clarified in the revision as per your valuable suggestion.
> >..weight transport problem is completely solved by random backprop..
>
> We agree that random BP is one plausible solution. We revised Sec. 1.1.2 to include "For example, the feedback alignment approach, which fixes randomly initialized feedback weights  and adapts feedforward weights, was offered as a plausible solution [17]". CorInfoMax is offered as an info theory based principled alternative hypothesis where networks of segregated neurons with recurrent and asymmetric feedback connections governed by local learning rules naturally emerge. Such a normative framework is useful in obtaining potential insights such as the role of lateral connections in embedding space expansion and avoiding degeneracy, feedback and feedforward connections for prediction to reduce redundancy, and activation functions/interneurons to shape feature space and compress. We will elaborate more on these in the revision by benefitting the extra page.
> >Questions
> >..not clear..at the top layer
>
> We utilize weak, not full, clamping based on the objective in (9a). As shown in Eq. (21) for the output layer, the error between the network output and labels influences network dynamics via the gain $\beta$.
> >"disputed" is too weak
>
> We agree. We will consider replacing the word "disputed" in the revised article.
> >"maximize linear dependence" why linear..identical activities..
>
> This is an important point to be clarified: typical network models use linear segments between layers (modelling synaptic integration), succeeded by nonlinear activations. In our framework, these linear segments in both directions emerge via correlation maximization. Additionally, set membership constraints, like the $\ell_1$-norm ball for sparsity, bring in nonlinearity. Thus, **linear mappings followed by nonlinear activtions** emerge from **CMI maximization under  domain set constraints for layers**.
> >"predictors" of what?
>
> To clarify, in the revised article we write "predictors of layer activation signals".
> >Fig. 1 is not easy to understand at this stage..
>
> We value your suggestion. Our aim is to maintain Fig.1 early on to offer a preview and motivate the discussion, if feasible.
> >..result by Liao..
>
> We will look into it more closely, thanks.
> >The term polytopic is not common..
>
> Polytopes, as compact intersections of half-spaces, allow flexible characterization of bounded layer activations. The choice of a polytope influences feature combinations like sparsity, antisparsity, and nonnegativity. Imposing these constraints leads to piecewise activations like ReLU and clipping functions, and introduces interneurons for sparsity. We will be happy to provide a brief summary as an appendix in the revision.
> >..stochastic in which sense?..specify..trying to maximize of minimize..
>
> First, we define the obj. function in ensemble average form using expectations. Later, we provide its sample average based form. We modified line 148 to specify the objective is "to be maximized".
> >Is there some Gaussian assumption..
>
> We make no Gaussian assumption. The correlative mutual information (CMI) is a second-order statistics-based measure, independent of the probability density functions, and it assesses the correlation level between its arguments. Please refer to Appendix A for an introduction to correlative entropy and CMI, which we expand in the revision.
> >..particularly unclear and hard to follow..
>
> We will modify Section 2 for better presentation and accessibility in the revised article.
> >Which problem?
>
> In the revision, we clarified the term "problem" to mean the optimization for finding the optimal linear regularized MMSE predictor, $\mathbf{W}\_{ff,\*}^{(k)}$, given in Eq. (3).
> >..no..limitations..
>
> We include a limitations section following your suggestion. Please see GRR item 5.

---

> ### Comment · Reviewer_wEH2 · 2023-08-22
>
> I agree that the reviewers have answered a fair number of comments--hard to tell something more precise without seeing the revised version. I am happy to move my score up by 1.

---

### Official Review · Reviewer_t54y · 2023-07-01

**Soundness:** 3 good
**Presentation:** 2 fair
**Contribution:** 3 good
**Rating:** 7
**Confidence:** 3

**Summary:**

The authors present a novel strategy for learning in neural networks. In particular, the authors derive update rules for neurons/synapses which maximises the correlative information between layer activations. This strategy avoids the weight transfer problem, and naturally gives rise to a biologically emulating architecture of multi-compartment pyramidal neurons with lateral inhibition.

**Strengths:**

- the authors present what seems a mathematically sound and creative strategy for credit assignment. Without extensive knowledge in this area, the derivation of update rules seems original and of good quality
- The resulting likeness to a multi-compartment model with lateral inhibition is interesting
- the text is generally well written (though the presentation itself is dense, see below)

**Weaknesses:**

- in general I found the paper very dense - I personally think 27 equations is too many for a main text. I appreciate that the main contribution of this paper is analytical, but the think the authors would do well to sacrifice some of the less key equations (move to SM) to make space for additional intepretation/experiments
- As stated above, I would have liked to have seen more intepretation and experiments with respect to the model. For example, what predictions does the model make in terms of the balance of bottom-up/top-down signals? How does this change over learning? How does it compare to biology? Same for interneurons
- The actual performance of the model does not seem too impressive, at least compared to standard backprop (e.g. on the CIFAR-10 dataset). Moreover, given that a key property of the model is to avoid the weight symmetry issue, I would think it sensible to compare the model to backprop with random feedback weights (feedback alignment).
- I think the authors coud make it more explicit what are the differences between their model and the model in Golkar et al. 2022. In particular, explicitly highlighting the similar and new terms when presenting the mathematical formulation

**Questions:**

- The segway at line 34 was unclear to me
- I found section 2.1 confusing to read because the equation which derives the activity r^k of a given layer is not expressed. Is this intentional to keep it general? It's confusing to understand whether the activities between the layers have a relationship at all at this point.
- line 140: the inequality used to express the hypercube is not well defined for vectors
- The CMI metric (equation 2) is a complicated equation with determinants and auto/cross-correlation matrices. I would have liked an intuitve
 (perhaps geometric) description of this measure
- line 150: R_{r^k r^{k+1}} hasn't appeared yet but is already being described
- in section 2.3.1 the variable s is introduced without any explanation as to what it represents. Is it time? i.e. u[t, s] is the t dataset example at time s?
- line 231: I didn't understand the decomposition of M. Where does D come from and why does it mirror autapses? Is it the identity matrix in the expression of M? WHere does the negative O part (interneuron) come from?
- equations 26,27: sorry if this is a naive question, but why can't the weights just be updated directly using equations 23,25?

**Limitations:**

I would recommend a limitations section (or at least more discussion). For example, I would be interested to know if the sensitivity of the model to hyperparameter choices is high, or whether there is a strict need for the feedback matrix at the last layer to be the identity.

---

> ### Author Rebuttal · Authors · 2023-08-08
>
> We greatly value your comprehensive review and constructive suggestions. While length constraints have necessitated brevity in our response, we anticipate the opportunity to discuss further details and address any outstanding queries during the discussion phase
>
> >..paper..dense..less key eq.s (move to SM)..
>
> Thanks. We followed your advise. Please see items 1 to 3 in global rebuttal response (GRR).
>
> >... more intepretation and experiments... what predictions does the model make..the balance of bottom-up/top-down signals?
>
> As noted in GRR-item 4, we've included new experiments.  As for the interpretations you've quoted, we believe our info. theory-based framework can provide insightful contributions and  we are currently focused on development. Although the balance of bottom-up and top-down signals presents an interesting research direction, we haven't researched this area yet. However, our principled approach offer potential insights such as the role of lateral connections in embedding space expansion and avoiding degeneracy, feedback and feedforward connections for prediction to reduce redundancy, and activation functions/interneurons to shape feature space and compress. We will elaborate more on these in the revision by benefitting the extra page.
> >..performance..not..too impressive..compared to standard BP..compare..to BP with random feedback..
>
> Following your suggestion, we performed additional experiments for comparisons with BP and feedback alignment for fully connected architecture. Updated Table 1 is available in Rebuttal pdf, which shows that CorInfoMax has on par performance with these benchmarks.
> >.. more explicit..differences between Golkar et al. ..highlighting...the mathematical formulation
>
> We can provide the following comparison between the CorInfoMax and the constrained predictive coding (C-PC) framework in Golkar et al.:
>
> * C-PC enhances existing maximum likelihood (ML) based predictive coding frameworks by integrating secondary forward prediction terms into the ML formulation. The minimization of negative log-likelihood can be expressed as:
> \begin{align}
> \min\_{\mathbf{Z},\mathbf{W}\_a,\mathbf{W}\_b} \hat{L}=\frac{1}{2}\sum_{l=1}^{n-1}\left[\frac{ \|\mathbf{Z}^{(l)}-\mathbf{W}^{(l-1)}_b\mathbf{Z}^{(l-1)}\|_F^2}{2{\sigma^{(l)}}^2}+\frac{ \|\mathbf{Z}^{(l+1)}-\mathbf{W}^{(l)}_a\mathbf{Z}^{(l)}\|_F^2}{2{\sigma^{(l+1)}}^2}\right].
> \end{align}
> Note that both terms in the summation are two separate forward prediction error terms.
>
> * In CorInfoMax, we propose maximization of correlative mutual information between sequential branches, where both forward and backward prediction matrices emerge from two alternative but equivalent forms of the CMI,
> * C-PC approach makes use of a whitening constraint on layer activations, which is utilized to convert forward prediction matrix $\mathbf{W}_a$ to a feedback matrix $\mathbf{W}_a^T$,
> * In CorInfoMax framework,  there is no whitening but a set membership constraint on layer activation vectors.
> * In C-PC lateral weights are based on Lagrangians of the covariance constraints,
> * In CorInfoMax lateral weights are inverse of the activation correlation matrix to maximize correlative entropy of activations,
> * The updates for feedforward and feedback matrices are different for both approaches (forward and backward prediction errors are used in the CorInfoMax).
>
> >Questions:
> >segway at line 34..unclear..
>
> The first paragraph addresses two main critiques concerning bio. plausibility: weight transport and simple neuron models. The subsequent paragraph delves into the weight symmetry issue, while the third one explores neuron models. We will ensure a more seamless transition between these.
>
> >sec. 2.1 confusing..the eq. which derives the activity $\mathbf{r}^k$..is not expressed. Is this intentional?
>
> Indeed, this is intentional. Equations reflecting activity, network structure, dynamics, and learning updates are not predetermined but they emerge from correlative information maximization with activation domain constraints.
>
> >line 140: the ineq..is not well defined
>
> Apologies for notation ambiguity. The ineq. $\mathbf{0} \leq \mathbf{r} \leq \mathbf{1}$ denotes elementwise comparison. We can change $\leq$ to $\preccurlyeq$.
>
> >The CMI metric (2) is a complicated..an intuitve (perhaps geometric) descrip. of this measure
>
> Thanks for the suggestion. We included the following in Sec. 2.2: *"If we interpret the maximization of CMI in (2): the first term on the right side of (2) encourages the spread of $\mathbf{r}^{(k+1)}$ in its presumed domain
> $\mathcal{P}^{(k+1)}$, while the second term incites the minimization of redundancy in $\mathbf{r}^{(k+1)}$  beyond its component predictable from $\mathbf{r}^{(k)}$."*
>
> >line 150: $\mathbf{R}_{\mathbf{r}^k \mathbf{r}^{k+1}}$ hasn't appeared yet
>
> Thanks. Removed. See GRR item 2.i.
>
> >in sec. 2.3.1..s is introduced without any explanation..
>
> Section 2.1 defines $t$ as the discrete data index, while $s$ is the continous time index for the network dynamics. We clarify this in the new appendix for network dynamics.
>
> >... the decomposition of M. Where does D come from ...
>
> We clarify it in the revision. Briefly, we define $\mathbf{D}^{(k)}$ as $\mathbf{D}^{(k)} = \text{diag}(\mathbf{M}^{(k)})$, i.e., a diagonal matrix containing the diagonal elements of $\mathbf{M}^{(k)}$. Then we define $\mathbf{O}^{(k)} = \mathbf{D}^{(k)} - \mathbf{M}^{(k)}$. Therefore, we can express $\mathbf{M}^{(k)}$ as $\mathbf{M}^{(k)} = \mathbf{D}^{(k)} - \mathbf{O}^{(k)}$.
>
> >..(26-27):..why can't the weights just be updated..using..23,25?
>
> Equilibrium Propagation updates in 26,27 target minimization of MSE error, whereas 23,25 are gradients for  the CMI based energy function determining system dynamics. The use of 23,25 still provides some accuracy but significantly below the EP updates.
>
> >Limitations:
> >I would recommend a limitations section ...
>
> Thanks, we include a limitations section following your suggestions. See GRR item 5.

---

> > ### Comment · Reviewer_t54y · 2023-08-12
> >
> > Thank you to the authors for their detailed response and explanations.
> >
> > Regarding the additional experiments, may I ask why feedback alignment with cross entropy loss is not included in the experiments presented in the new Tables 1/2 (whilst cross entropy loss is used with standard BP)?

---

> > > ### Author Response · Authors · 2023-08-12
> > > **Response to Reviewer Comment**
> > >
> > > Thank you again for your comments and questions. We considered MSE loss appropriate for the feedback-alignment experiments, especially since we employ MSE loss within the CorInfoMax framework. However, we will be happy to incorporate feedback-alignment experiments utilizing cross-entropy loss. We have initiated these experiments and will share the results once they are available.

---

> > > > ### Author Response · Authors · 2023-08-15
> > > > **Response to Reviewer Comment - Continued**
> > > >
> > > > We have completed our experiments for Feedback Alignment with the Cross-Entropy loss. Below, you can find the updated results in Table 1 for 2-layer networks and Table 2 for 3-layer networks.
> > > >
> > > >
> > > > **Table 1: 2-layered Neural Network Results**
> > > >
> > > > Algorithm \ Dataset |      MNIST    |    Fashion-MNIST | CIFAR10
> > > > :--------------------:|:--------------------------------:|:--------------------:|:-------------------------:
> > > > CorInfoMax-$\mathcal{B}_{\infty, +}$ | 97.62 $\pm$ 0.1 | 88.14 $\pm$ 0.3 | 51.86 $\pm$ 0.3|
> > > > CorInfoMax-$\mathcal{B}_{1, +}$ | 97.71 $\pm$ 0.1 | 88.09 $\pm$ 0.1 | 51.19 $\pm$ 0.4|
> > > > EP | 97.61 $\pm$ 0.1| 88.06 $\pm$ 0.7 | 49.28 $\pm$ 0.5 |
> > > > CSM | 98.08 $\pm$ 0.1| 88.73 $\pm$ 0.2 | 40.79 |
> > > > PC | 98.17 $\pm$ 0.2 | 89.31 $\pm$ 0.4 | - |
> > > > PC-Nudge | 97.71 $\pm$ 0.1| 88.49 $\pm$ 0.3 | 48.58 $\pm$ 0.7|
> > > > Feedback-Alignment (with MSE) | 97.99 $\pm$ 0.03| 88.72 $\pm$ 0.5| 50.75 $\pm$ 0.4 |
> > > > Feedback-Alignment (with CrossEntropy)|  97.95 $\pm$ 0.08 | 87.38 $\pm$ 0.92 | 52.37 $\pm$ 0.38 |
> > > > BP (with MSE) |97.58 $\pm$ 0.01 |88.39 $\pm$ 0.1 |52.75 $\pm$ 0.1 |
> > > > BP (with CrossEntropy) | 98.27 $\pm$ 0.03| 89.41 $\pm$ 0.2| 53.96 $\pm$ 0.3 |
> > > >
> > > >
> > > > **Table 2: 3-layered Neural Network Results**
> > > >
> > > > Algorithm \ Dataset | MNIST | CIFAR10 | CIFAR100 |
> > > > :--------------------:|:-------------------------:|:--------------------:|:---------------------------|
> > > > CorInfoMax-$\mathcal{B}_{\infty, +}$ | 97.58 $\pm$ 0.1 | 50.97 $\pm$ 0.4 | 20.84 $\pm$ 0.4 |
> > > > Feedback-Alignment (with MSE) | 98.18 $\pm$ 0.0 | 50.26 $\pm$ 1.4| - |
> > > > Feedback-Alignment (with CrossEntropy)|  97.96 $\pm$ 0.15 | 51.64 $\pm$ 0.60 | - |
> > > > BP (with MSE) | 97.74 $\pm$ 0.0 | 55.49 $\pm$ 0.04 | 26.56 $\pm$ 0.2 |
> > > > BP (with CrossEntropy) | 98.28 $\pm$ 0.04 | 56.14 $\pm$ 0.3 | 28.93 $\pm$ 0.3 |

---

> > > > > ### Comment · Reviewer_t54y · 2023-08-15
> > > > >
> > > > > Thank you to the authors for their new simulations.
> > > > >
> > > > > On one hand I am a bit disappointed to see that CorInfoMax does not improve upon standard FA for multi-layer networks, but on the other hand the authors have convinced me that the work might still be a very valuable contribution for neuroscience and bio-plausible learning. I also appreciate the great deal of work this study must have taken and the authors' commitment to rigorous explanations in their rebuttal response.
> > > > >
> > > > > I will upgrade my score by 1.

---

> > > > > > ### Author Response · Authors · 2023-08-15
> > > > > > **Response to Reviewer Comment**
> > > > > >
> > > > > > We want to extend our thanks once again for your useful comments and engaging questions. Your thoughtful considerations have allowed us to refine our method even further, and we are truly thankful for your positive evaluation.

---

### Official Review · Reviewer_K4kq · 2023-07-05

**Soundness:** 3 good
**Presentation:** 2 fair
**Contribution:** 3 good
**Rating:** 6
**Confidence:** 4

**Summary:**

The authors introduce a biologically plausible training paradigm for a deep neural network that sidesteps the weight transport problem while achieving competitive results. Their approach is normative, in that both the network's architecture as well as its learning rules can be derived from an information maximization approach. The asymmetry between forward and backward weights is achieved by leveraging two different formulations of the inter-layer correlative information.

**Strengths:**

*Originality:*
The work provides a novel approach for deriving biologically plausible strategies for learning in deep neural networks.

*Quality:*
The paper contains a significant amount of work to support its findings. Importantly, both theory and computation are used in tandem.

*Clarity:*
The presentation is mostly clear, but some significant explanations are missing or too sparse. See below. I also want to praise the authors for including the code that they used with the submission (something that I believe should be true for all papers, but sadly is not).

*Significance:*
The work is significant for neuroscience because the learning algorithms used by the brain are not yet understood. Having a good grasp over the range of possible mechanisms that biology could have used to train natural neural networks is essential to allow experimentalists to probe what choice(s) is (are) actually used. The work is also of potential significance for machine learning, since the algorithms used by the brain might provide advantages over the gradient descent with backpropagation methods used to train artificial neural networks.


**Weaknesses:**

1. The correlative mutual information metric requires a bit more discussion. The regularization coefficients $\epsilon_k$ appear in eq. (2) but are not discussed at all until much later, and even in the derivation in Appendix A, the need for this regularization is not explained. On first guess, the need for $\epsilon_k \ne 0$ is due to having a low rank covariance matrix $\mathbf R_{\mathbf r^{(k+1)}}$. However, this seems inconsistent with the importance of these coefficients in the network dynamics and learning rules. This requires a more detailed discussion in the main text, and especially in the Appendix. (If space is an issue, I suggest removing most of lines 170-174, which are almost identical to 149-153; it can simply be stated that the sample covariance matrices from eq. (6) need to be used instead of their exact counterparts to get online training rules.)
2. Related to the regularization coefficients, I am a bit perplexed by eqns. (10), (11). The Taylor expansion in these equations is performed around the identity, but that makes the expansion parameter be $1 / \epsilon$. Since $\epsilon$ is small, $1 / \epsilon$ should be big, making it hard to justify ignoring subsequent terms in the Taylor expansion. This needs to be explained in detail.
3. The jump to the dynamics equations (15)–(17) is too abrupt. Either an explanation should be provided or a reference to a relevant Appendix section.

**Questions:**

Questions
* for me, phrasing the method in terms of using two *equivalent* forms of the correlative mutual information is misleading: if they are equivalent, how could they lead to the desired asymmetry between feed-forward and feed-back weights? I may be wrong, but I believe that the resolution lies in the Taylor expansion around line 178 -- it's not the differing exact expressions that lead to the asymmetry, but the different *approximations*
* the definition of $\mathbf M$ on line 212 is a bit confusing. Should the factor of 2 apply to both terms? I would have thought that the leak term in the expression for $\mathbf M$ is supposed to counteract the leak term in eq. (15) as long as $\mathbf u$ is inside the feasible domain
* also regarding $\mathbf M$: since $\epsilon$ is small, my understanding is that $\mathbf M$ is also small; in this case, I would suggest leaving $\epsilon$ out of the definition for $\mathbf M$, to make the scale of each term more apparent; e.g., in eq. (18), I would imagine $\mathbf W_{fb}$ has the leading contribution to $\mathbf v_A$, while the $\mathbf M$ term is a sub-leading correction
* were hyperparameters optimized for each task for all of the algorithms in Table 1?

Minor comments:
* in eq. (12), the notation $\hat J_k$ is used but it had never been introduced before. Please define $\hat J_k$ first before using
* the update rule for the lateral coefficients (lines 280-281) should be written in terms of the biologically motivated parameters ($\mathbf M$, or even better, $\mathbf D$ and $\mathbf O$) instead of $\mathbf B$.
* in the numerical experiments section, please include at least basic details about the networks that are used (e.g., number of layers)
* below eq. (A.12), the clipping operation $\sigma_+$ is invoked out of nowhere. I think the point is to show how this operation can be justified as a way of enforcing the KKT conditions; this should be made clearer
* the plots in Appendix Figures 7–8, 12–14 are very sparse and it's not clear what information we are to glean from them; I suggest combining some of these, e.g., compare the accuracy attained by different network architectures and / or activation functions
* in the studies from Appendix F, the range over which the hyperparameters are varied seems too small because the variation in accuracy is very low, barely beyond the variability over different runs; I suggest using larger variations to show the trend

**Limitations:**

The authors have adequately discussed limitations of their work.

---

> ### Author Rebuttal · Authors · 2023-08-08
>
> We thank you for your detailed review and constructive comments. Due to the strict length constraint some details we prepared were needed to be removed. We will be happy to provide more details and answer your potential questions during the discussion period.
> >Strengths:
>
> We sincerely value your recognition of our work's novelty and significance
> > Weaknesses:
> >..Discussion on CMI metric, and regularization parameter $\epsilon_k$..
>
> Thanks a lot for this constructive suggestion. Please see items 2 - 3 in our global rebuttal response.
> >..Taylor exp. in (10-11) is performed around the identity..
>
> Our revised article includes an appendix section detailing the linearization based on the truncated Taylor series. Briefly, the linearization around $\mathbf{A}$ with perturbation $\mathbf{\Delta}$ can be expressed as
> $\log\det(\mathbf{A}+\mathbf{\Delta})\approx \log\det(\mathbf{A})+\text{Tr}(\mathbf{A}^{-1}\mathbf{\Delta}).$
> For the correlative entropy of the prediction error,  $\log\det(\epsilon \mathbf{I}+\mathbf{R}\_\mathbf{e})$, we assume $\epsilon \mathbf{I} \succ \mathbf{R}\_\mathbf{e}$ and  linearize around $\mathbf{A}=\epsilon \mathbf{I}$ with perturbation $\mathbf{\Delta}=\mathbf{R}\_\mathbf{e}$, yielding $\log\det(\epsilon \mathbf{I}+\mathbf{R}\_\mathbf{e})\approx \log\det(\epsilon \mathbf{I})+\epsilon^{-1}\text{Tr}({\mathbf{R}\_\mathbf{e}})$.
>  In summary, the perturbation term is not $\epsilon \mathbf{I}$ but ${\mathbf{R}\_\mathbf{e}}$. The assumption follows from the discussion on the impact of $\epsilon$, where maximizing the CMI is achieved by pushing the eigenvalues of ${\mathbf{R}_\mathbf{e}}$ below $\epsilon$, for *reasonable* values of $\epsilon$. For our nominal choice of $\epsilon=0.15$ in our experiments, it is indeed the case.
> >..jump to the dynamics equations (15-17) is too abrupt..
>
> Thanks again for this suggestion. We  include a new appendix section on the more detailed derivation of (15-17).
> >Questions:
> >phrasing the method in terms of using two equivalent forms of the CMI is misleading: .. how could they lead to the desired asymmetry between feed-forward and feed-back weights? ..it's not the differing exact expressions that lead to the asymmetry, but the different approximations?
>
> Briefly, asymmetry is **not** due to Taylor series based approximation: Equations (7) and (8) indeed represent two equivalent alternatives for  Correlative Mutual Information (CMI), but individual components vary. Specifically,   the first terms of (7) and (8) represent the Correlative Entropy (CE) of layer $k+1$ and layer $k$ activations, while the second terms are the CEs of forward and backward prediction errors. These are not necessarily equivalent,  leading to inherently unequal forward and backward error entropies and corresponding weight matrices. Taylor series based approximation is just for the linearization of the forward and backward prediction entropies. As discussed in Appendix B.2, we can write the forward and backward predictor weights as
> $\mathbf{W}\_{ff,\*}^{(k)}=\mathbf{R}\_{\mathbf{r}^{(k+1)}\mathbf{r}^{(k)}}(\mathbf{R}\_{\mathbf{r}^{(k)}}+\epsilon_k \mathbf{I})^{-1},$
> $\mathbf{W}\_{fb,\*}^{(k)}=\mathbf{R}\_{\mathbf{r}^{(k)}\mathbf{r}^{(k+1)}}(\mathbf{R}\_{\mathbf{r}^{(k+1)}}+\epsilon_k \mathbf{I})^{-1}$
>  $(\mathbf{R}\_{\mathbf{r}^{(k)}}+\epsilon_k \mathbf{I})^{-1}$ and $(\mathbf{R}\_{\mathbf{r}^{(k+1)}}+\epsilon_k \mathbf{I})^{-1}$. Consequently, the condition $\mathbf{W}\_{ff}^{(k)}={\mathbf{W}\_{fb}^{(k)}}^T$ does not generally hold true. Symmetry might be anticipated in very specific scenarios - such as diagonal  autocorrelation matrices. This analysis only considers the mutual information maximization component of the objective, yet it offers insight into the expected asymmetry of the forward and backward weights.
> >..the definition of $\mathbf{M}$.. the factor of 2 apply to both terms?
>
> Thanks for pointing the typo. We fixed as $\mathbf{M}^{(k)}[t] = \epsilon_k(2 \gamma\mathbf{B}^{(k)}[t] + g_{\text{lk}} \mathbf{I}$)
> >... $\mathbf{M}$ is also small ... I would suggest leaving $\mathbf{\epsilon}$ out of the definition...
>
> Thanks. The goal was to obtain compact representation. We will evaluate your suggestion for the revision.
> >were hyperparameters optimized for each task for all of the algorithms in Table 1
>
> Indeed, we put considerable effort into optimizing the hyperparameters for most of the tasks and all the algorithms in Table 1, using grid search. Our shared code explicitly include the grid search parameters associated with each algorithm. Additionally, our Python notebooks under "AnalyzeSimulations" present the train and test results in a comprehensive table, enabling easy comparison between various settings.
> >Minor Comments:
> >in eq. (12), ..$\mathbf{\hat{J}_k}$.. had never been introduced..
>
> Thanks. In the revision,  we explicity include ${\hat{J}}\_k(\mathbf{r}^{(k)}) = \overset{\rightarrow}{\hat{I}^{(\epsilon\_{k-1})}}(\mathbf{r}^{(k - 1)},\mathbf{r}^{(k)})[t]+\overset{\leftarrow}{\hat{I}^{(\epsilon\_k)}}(\mathbf{r}^{(k)},\mathbf{r}^{(k+1)})[t]$ for $k=1, \ldots, P$ and $\hat{J}\_P(\mathbf{r}^{(P)})[t]=\overset{\rightarrow}{\hat{I}^{(\epsilon\_{P-1})}}(\mathbf{r}^{(P - 1)},\mathbf{r}^{(P)})[t]-\frac{\beta}{2}\|\mathbf{r}^{(P)}[t]-\mathbf{y}\_T[t]\|_2^2$.
> >the update rule for the lateral coefficients ... in terms of ... ($\mathbf{M}$, or ..., $\mathbf{D}$ and $\mathbf{O}$)
>
> We'll revise the update equations as suggested (see Alg. 1 in global PDF).
> >... experiments section...include...details about the networks..
>
> Thanks. We provide them in the appendix but we will include in the main text of the revision.
> >below (A.12)...$\sigma_+$ is invoked out of nowhere ..
>
> We reworded  this sentence.
> >... Figures 7–8, 12–14 are very sparse...suggest combining...
>
> Following your advice, we merged 7-9 and 12-14.
> >Appendix F...hyperparameters...suggest using larger variations to show the trend...
>
> Revised manuscript will include an extended ablation study section in appendix.

---

> > ### Comment · Reviewer_K4kq · 2023-08-11
> >
> > Thanks for the explanations.
> >
> > It seems like the answer regarding the $\epsilon$ terms is that they are actually relatively *large*, not small. I suppose this is a valid choice, although I'd be curious to know whether the quadratic term in the Taylor expansion is actually negligible compared to the linear one that you keep.
> >
> > Regardless of this, such a large value for $\epsilon$ goes well beyond regularization in the sense of avoiding divergences, it appears to act more like a prior, pushing your system in a direction that presumably is desirable for some reason. What is that direction and why do you want to bias the system like this?
> >
> > In other words, by using a large $\epsilon$, you are not quite optimizing correlative mutual information anymore, and in that case, you should explain what you are optimizing instead, and why.

---

> > > ### Author Response · Authors · 2023-08-12
> > > **Response to Reviewer Comment - Part 1**
> > >
> > > Thank you for your comments and questions, which enhance the understanding of our article. Before directly addressing your query, we will first delve deeper into the discussion on the $\epsilon$ parameter, and then offer our responses:
> > >
> > > Upon examining the expression for correlative mutual information (CMI),
> > > $$\overset{\rightarrow}{{I}^{(\epsilon_k)}}(\mathbf{r}^{(k)}, \mathbf{r}^{(k+1)}) = \frac{1}{2} \log \det \left(\mathbf{R}\_{\mathbf{r}^{(k+1)}} + \epsilon_k \mathbf{I}\right)- \frac{1}{2} \log \det \left(\mathbf{R}\_{\overset{\rightarrow}{\mathbf{e}^{(k+1)}\_\*}} + \epsilon_k \mathbf{I}\right), \quad (2)$$
> > > $\epsilon_k$ appears to function as a correction factor to compensate for rank-deficient correlation matrices of degenerate random vectors. From this perspective, this adjustment serves two primary purposes:
> > >  * To establish a finite lower bound for the entropy, and
> > >  * To circumvent numerical optimization issues, given that the derivative of the $\log\det$ function is the inverse of its argument.
> > > In fact, robust matrix factorization methods that rely on determinant-maximization use this perturbation for aforementioned reasons [a]. Additionally, a recent study links $\epsilon_k$ with a parameter of the **Inverse Wishart prior** distribution on the covariance matrix of the row vectors of one of the factors [b].
> > >
> > > Moving beyond these interpretations, we first observe that (2) defines **a family** of correlative mutual information definitions. **For each $\epsilon_k$ choice, we have a valid alternative correlative mutual information definition [35]. Hence, we do not necessarily interpret them as approximations of $\epsilon_k=0$ case, instead they are alternative CMI measures.** To see the impact of $\epsilon_k$ choice:
> > > -We note that the prediction error covariance matrix in (2) $$\mathbf{R}\_{\mathbf{e}^{(k+1)}\_\*}=\mathbf{R}\_\mathbf{r^{(k+1)}} - \mathbf{R}\_{\mathbf{r}^{(k)} \mathbf{r}^{(k+1)}}^T(\mathbf{R}\_\mathbf{r^{(k)}} + \epsilon_k \mathbf{I})^{-1} \mathbf{R}\_{\mathbf{r^{(k)}}\mathbf{r}^{(k+1)}} \hspace{0.2in}(A)$$ corresponds to the error correlation matrix for the best linear regularized minimum mean square estimator of $\mathbf{r}^{(k)}$ from $\mathbf{r}^{(k+1)}$. This estimator is obtained as  the solution of the optimization problem
> > > \begin{eqnarray}
> > >     \underset{\mathbf{W}\_{\mathbf{r}^{(k+1)}|\mathbf{r}^{(k)}}}{\text{minimize }} {E(\||{\mathbf{r}^{(k+1)}}-\mathbf{W}\_{\mathbf{r}^{(k+1)}|\mathbf{r}^{(k)}}\mathbf{r}^{(k)}\||\_2^2)+\epsilon_k\||\mathbf{W}\_{\mathbf{r}^{(k+1)}|\mathbf{r}^{(k)}}\||\_F^2}.
> > > \end{eqnarray}
> > > In this context, $\epsilon_k$ parameter acts as a regularizing coefficient for the linear estimation problem integral to measuring linear dependence between the two arguments of the CMI.
> > > Maximizing the CMI given by equation (2) can be accomplished by increasing the correlative entropy of  $\mathbf{r}^{(k+1)}$ while decreasing the correlative entropy of the estimation error $\mathbf{e}^{(k+1)}\_\*$. Based on Eq. (A) above, we have $\mathbf{R}\_{\mathbf{r}^{(k+1)}}\succeq \mathbf{R}\_{\mathbf{e}^{(k+1)}\_\*}$, and given that we can write the CMI in (2) as,
> > > $$\overset{\rightarrow}{{I}^{(\epsilon_k)}}(\mathbf{r}^{(k)}, \mathbf{r}^{(k+1)})=\frac{1}{2}\sum_{l=1}^{N_{k+1}}(\log(\lambda_l(\mathbf{R}\_{\mathbf{r}^{(k+1)}})+\epsilon_k)-\log(\lambda_l(\mathbf{R}\_{\mathbf{e}^{(k+1)}\_\*})+\epsilon_k))$$
> > > where $\lambda_l(\mathbf{R})$ denotes the $l$-th eigenvalue of the matrix $\mathbf{R}$.
> > >  We anticipate that the choice of $\epsilon_k$  will primarily influence the correlative entropy of $\mathbf{e}^{(k+1)}\_\*$. Indeed, since $\epsilon_k$ is added to all the eigenvalues of $\mathbf{R}\_{\mathbf{e}^{(k+1)}\_\*}$, reducing its eigenvalues below $\epsilon_k$ would have only incremental increase in the mutual information. As such, a smaller $\epsilon$ value will place greater emphasis on reducing the estimation error  $\mathbf{e}^{(k+1)}\_\*$. Consequently, one can consider  $\epsilon^{-1}$  can be viewed as an indicator of the sensitivity of the CMI to the levels of estimation error $\mathbf{e}^{(k+1)}\_\*$ (hence act as a conductance for  basal/appical-soma connections), determining how far we need to push down the estimation error values to increase the CMI.
> > >
> > > In brief, the choice of $\epsilon_k$ refers to choice of a CMI from a family of CMIs. Furthermore, the choice of $\epsilon_k$ determines the relative contributions of first and second terms in (2), where smaller $\epsilon_k$ (or larger $\epsilon_k^{-1}$) choice give more emphasis to prediction error entropy.
> > >
> > >
> > > [a] Xiao Fu et. al. Robust volume minimization-based matrix factorization for remote sensing and document clustering. IEEE TSP, Aug 2016.
> > >
> > >
> > > [b] Tatli G et. al. A Bayesian Perspective for Determinant Minimization Based Robust Structured Matrix Factorization. In ICASSP 2023 Jun 2023.

---

> > > > ### Author Response · Authors · 2023-08-12
> > > > **Response to Reviewer Comment - Part 2**
> > > >
> > > > In the light of these explanations, we can provide following answers to your questions:
> > > >
> > > > >It seems like the answer regarding the
> > > >  terms is that they are actually relatively large, not small. I suppose this is a valid choice, although I'd be curious to know whether the quadratic term in the Taylor expansion is actually negligible compared to the linear one that you keep.
> > > >
> > > > The use of a Taylor series-based approximation allows us to transform the nonlinear function of error covariance into an affine form. This transformation facilitates a more convenient network implementation. We acknowledge that there might be inaccuracies in this approximation with potentially a high margin of error, especially depending on the evolution of prediction error correlation matrix. The direct use of the error entropy or alternative approximation strategies might be considered as potential future extensions of our framework.
> > > >
> > > > >Regardless of this, such a large value for
> > > >  goes well beyond regularization in the sense of avoiding divergences, it appears to act more like a prior, pushing your system in a direction that presumably is desirable for some reason. What is that direction and why do you want to bias the system like this?
> > > >
> > > >  As explained above, the choice of $\epsilon_k$ essentially determines the relative contributions of embedding and prediction error correlation matrices on the CMI. Lower $\epsilon_k$ implies giving more emphasis to prediction error related component of the CMI. This fact is more evident in the linear approximation based version of the CMI:
> > > >
> > > >  $$\overset{\rightarrow}{{I}^{(\epsilon_k)}}(\mathbf{r}^{(k)}, \mathbf{r}^{(k+1)}) \approx \frac{1}{2} \log \det \left(\mathbf{R}\_{\mathbf{r}^{(k+1)}}\right)- \frac{\epsilon_k^{-1}}{2} \text{Tr} \left(\mathbf{R}\_{\overset{\rightarrow}{\mathbf{e}^{(k+1)}\_\*}} \right)+const$$
> > > >
> > > >  where $\epsilon_k^{-1}$ is the weight of prediction error covariance dependent term. This is also linked to the fact that $\epsilon_k^{-1}$ acts as a conductance term for incorporating forward and backward prediction errors.
> > > >
> > > >
> > > >
> > > > > In other words, by using a large $\epsilon$
> > > > , you are not quite optimizing correlative mutual information anymore, and in that case, you should explain what you are optimizing instead, and why.
> > > >
> > > >   Regardless of the choice for  $\epsilon_k$, the associated $\overset{\rightarrow}{{I}^{(\epsilon_k)}}(\mathbf{r}^{(k)}, \mathbf{r}^{(k+1)})$ is a legitimate  CMI. Thus, we are optimizing a CMI selection from a set of CMIs. The choice of $\epsilon_k$, and therefore, the choice of the corresponding CMI, determine  the relative influence of the correlation matrices of the embeddings and prediction errors. It is important to note  that these $\epsilon_k$ values are neural network hyperparameters and can be adjusted for  better accuracy of the network.

---

> > > > > ### Comment · Reviewer_K4kq · 2023-08-12
> > > > >
> > > > > Thanks for the explanations. I think at this point it's just a question of semantics: sure, you can think of each value of $\epsilon$ as defining a new CMI, but the larger that value, the farther that quantity becomes from the concept of mutual information that's it's based on and named after. I think that's fine as long as this is clearly explained in the text. I also think it takes away a bit of the claim of a purely normative approach, but it's not much.

---

> > > > > > ### Author Response · Authors · 2023-08-13
> > > > > > **Response to Reviewer Comment**
> > > > > >
> > > > > > We deeply appreciate your comment and recognize the nuances of the topic at hand. To ensure there's no ambiguity, please allow us to offer further clarity on this matter:
> > > > > >
> > > > > > **TLDR:** You've correctly pointed out that as $\epsilon$ increases , the CMI-${{I}^{(\epsilon)}}(\mathbf{x}, \mathbf{y})$- deviates from the form of Shannon Mutual Information associated with two jointly Gaussian vectors. However, it is still a legitimate *correlative* mutual information measure that reflect correlation (not necessarily dependence) between two random vectors.
> > > > > >
> > > > > > We draw a comparison between Shannon Mutual Information and Correlative Mutual Information to clarify our stance:
> > > > > >
> > > > > > **Shannon Mutual Information**
> > > > > > Shannon Mutual Information (SMI), $I_{SMI}(\mathbf{x},\mathbf{y})$ between two random vectors $\mathbf{x}$ and $\mathbf{y}$, we can list the following properties:
> > > > > > - Its computation typically requires  full joint pdf $f_{\mathbf{x},\mathbf{y}}$,
> > > > > > - It is nonnegative, i.e., $I_{SMI}(\mathbf{x},\mathbf{y})\ge 0$,
> > > > > > - It is zero,i.e., $I_{SMI}(\mathbf{x},\mathbf{y})=0$ if and only if $\mathbf{x}$ and $\mathbf{y}$ are **independent**,
> > > > > > - Due to the previous property, $I_{SMI}$ is a measure of **dependence** between two vectors,
> > > > > > - If $\mathbf{x}$ and $\mathbf{y}$ are zero-mean jointly Gaussian non degenerate vectors, i.e., $\left[\begin{array}{c}\mathbf{x}\\\ \mathbf{y}\end{array}\right]\sim \mathcal{N}\left(\mathbf{0},\left[\begin{array}{cc} \mathbf{R}\_\mathbf{x} & \mathbf{R}\_{\mathbf{x}\mathbf{y}} \\\ \mathbf{R}\_{\mathbf{y}\mathbf{x}} &\mathbf{R}\_\mathbf{y}\end{array}\right]\right)$, then
> > > > > > $$I_{SMI}(\mathbf{x},\mathbf{y})=\frac{1}{2}\log\det(\mathbf{R}\_\mathbf{x})-\frac{1}{2}\log\det(\mathbf{R}\_{\mathbf{e}\_{\mathbf{x}|\mathbf{y}}}),$$ where $\mathbf{R}\_{\mathbf{e}\_{\mathbf{x}|\mathbf{y}}}=\mathbf{R}\_\mathbf{x}-\mathbf{R}\_{\mathbf{x}\mathbf{y}}\mathbf{R}\_{\mathbf{y}}^{-1}\mathbf{R}\_{\mathbf{y}\mathbf{x}}$ is the error correlation matrix of the best linear minimum mean square (MMSE) estimator of $\mathbf{x}$ from $\mathbf{y}$.
> > > > > >
> > > > > >
> > > > > > **Correlative Mutual Information**
> > > > > > Correlative Mutual Information (CMI) with parameter $\epsilon \ge 0$, $I^{(\epsilon)}(\mathbf{x},\mathbf{y})$ between two random vectors $\mathbf{x}$ and $\mathbf{y}$, which is defined as
> > > > > >
> > > > > > $$I^{(\epsilon)}(\mathbf{x},\mathbf{y})=\frac{1}{2}\log\det(\mathbf{R}\_\mathbf{x}+\epsilon\mathbf{I})-\frac{1}{2}\log\det(\mathbf{R}\_{\mathbf{e}\_{\mathbf{x}|\mathbf{y}}}+\epsilon\mathbf{I}),$$
> > > > > >
> > > > > > where $\mathbf{R}\_{\mathbf{e}\_{\mathbf{x}|\mathbf{y}}}=\mathbf{R}\_\mathbf{x}-\mathbf{R}\_{\mathbf{x}\mathbf{y}}(\mathbf{R}\_{\mathbf{y}}+\epsilon \mathbf{I})^{-1}\mathbf{R}\_{\mathbf{y}\mathbf{x}}$ is the error correlation matrix of the best $\epsilon$-regularized  linear MMSE estimator of $\mathbf{x}$ from $\mathbf{y}$.
> > > > > >
> > > > > > we can list the following properties for the CMI:
> > > > > >
> > > > > > - Its computation only requires second order statistics, i.e., $\mathbf{R}\_\mathbf{x}$, $\mathbf{R}\_{\mathbf{x}\mathbf{y}}$ and $\mathbf{R}\_\mathbf{y}$, irrespective of the joint pdf $f_{\mathbf{x}\mathbf{y}}$,
> > > > > > - It is always nonnegative, i.e., $I^{(\epsilon)}(\mathbf{x},\mathbf{y})\ge 0$.
> > > > > > - It is zero,i.e., $I^{(\epsilon)}(\mathbf{x},\mathbf{y})=0$ if and only if $\mathbf{x}$ and $\mathbf{y}$ are **uncorrelated**,
> > > > > > - Due to the previous property, $I^{(\epsilon)}$ is a measure of **correlation** between two vectors,
> > > > > > - For the choice $\epsilon=0$, $I^{(0)}(\mathbf{x},\mathbf{y})$ is equal to $I_{SMI}$ of two jointly Gaussian vectors with the same second order statistics.
> > > > > >
> > > > > > As a result, for the CMI $I^{(\epsilon)}(\mathbf{x},\mathbf{y})$ with a choice of parameter $\epsilon>0$ (stricly positive), we can make the following observations:
> > > > > > - ${I}^{(\epsilon)}$ is not equal to the form of $I_{SMI}$ betweeen two Gaussian vectors,
> > > > > > - Therefore, it is not a measure of dependence,
> > > > > > - However, it is still a legitimate measure of correlation between two random vectors, i.e., $I^{(\epsilon)}(\mathbf{x},\mathbf{y})\ge 0$ with equality if and only if $\mathbf{x}$ and $\mathbf{y}$ are uncorrelated (Lemma 1 of [35]).

---

> > > > > > > ### Comment · Reviewer_K4kq · 2023-08-13
> > > > > > >
> > > > > > > If I understand correctly, the argument is that we are more interested in the fact that the CMI is a measure of correlation between two vectors. But there are many possible measures of correlation—simplest one being the correlation itself. Maximizing correlation leads to CCA, as for example Lishutz, Bahroun, Golkar et al. did in "A Biologically Plausible Neural Network for Multichannel Canonical Correlation Analysis", *Neural Computation* (2021). How different is your approach compared to (a nonlinear version of) that?
> > > > > > >
> > > > > > > Of course, I'm not suggesting you should change your approach at this point! I'm just saying that there were many choices made along the way that led to your method as opposed to something else, and I think these should be acknowledged better.

---

> > > > > > > > ### Author Response · Authors · 2023-08-14
> > > > > > > > **Response to Reviewer Comment**
> > > > > > > >
> > > > > > > > Thank you for your insightful comment and request for clarification. Let's delve deeper into the relationship between Canonical Correlation Analysis (CCA) and Correlative Mutual Information (CMI), as well as how reference [a] relates to our framework:
> > > > > > > >
> > > > > > > > **Canonical Correlation Analysis**
> > > > > > > >
> > > > > > > > Given two zero-mean random vectors $\mathbf{x}\in \mathbb{R}^m$ and $\mathbf{y}\in \mathbb{R}^n$, CCA finds  projection matrices $\mathbf{V}\_\mathbf{x}\in \mathbb{R}^{p \times m}$ and $\mathbf{V}\_\mathbf{y}\in \mathbb{R}^{p \times n}$, where $p=\text{min}(m,n)$, such that the pairwise correlations of the elements of projection vectors $\mathbf{q}=\mathbf{V}\_\mathbf{x}\mathbf{x}$ and $\mathbf{w}=\mathbf{V}\_\mathbf{y}\mathbf{y}$ are maximized.  Specifically, the aim is to maximize the sum of the correlations  $\rho_i=E(q_iw_i)$, while maintaining unity-variance output constraints on the projection matrices.
> > > > > > > >
> > > > > > > > **Connection between CCA and CMI**
> > > > > > > >
> > > > > > > > As per reference [b], when $\mathbf{x}$ and $\mathbf{y}$ are jointly Gaussian, Shannon Mutual Information of $\mathbf{x},\mathbf{y}$ (and consequently  the CMI with parameter $\epsilon=0$) can be expressed in terms of cannonical correlations as:
> > > > > > > >
> > > > > > > > $$ I^{(0)}(\mathbf{x},\mathbf{y})=-\frac{1}{2}\sum_{i=1}^p\log(1-\rho_i^2)\hspace{0.2in} (*)$$
> > > > > > > >
> > > > > > > > Thus, while CCA can be employed to compute $I^{(0)}(\mathbf{x},\mathbf{y})$, it doesn't inherently maximize it.
> > > > > > > >
> > > > > > > > **About the biologically plausible CCA approach of [a]**
> > > > > > > >
> > > > > > > > In essence, reference [a] introduces a framework for biologically plausible neural networks that produce projections $\mathbf{q}$ and $\mathbf{w}$ with optimal canonical correlations from inputs $\mathbf{x}$ and $\mathbf{y}$. For the biologically plausible CCA network of [a], we can list the following:
> > > > > > > > - the samples of $\mathbf{x}$ and $\mathbf{y}$ serve as strict inputs; they are not manipulated by the CCA network,
> > > > > > > > - the network computes the projections $\mathbf{q}$ and $\mathbf{w}$ of these inputs, which have maximum sum of elementwise correlations.
> > > > > > > > - Overall the network calculates CCA components corresponding to its inputs as a result of a dynamical optimization process. However, this network does not manipulate $\mathbf{x}$ and $\mathbf{y}$ to maximize their canonical correlations,
> > > > > > > > - The element-wise correlations of the outputs of the CCA network correspond to canonical correlations $\rho_i$, which can then be used to determine the CMI $I^{(0)}(\mathbf{x},\mathbf{y})$.
> > > > > > > >
> > > > > > > >
> > > > > > > > **The relationship of CorInfoMax framework and the reference [a]**
> > > > > > > >
> > > > > > > > As described above,
> > > > > > > > - the biologically plausible CCA networks of Reference [a]  can be used to calculate CCA components corresponding the inputs $\mathbf{x}$ and $\mathbf{y}$. As such, it can also be employed to compute the CMI $I^{(0)}(\mathbf{x},\mathbf{y})$ using Eq. (*) above if extended. However, the CCA network in [a] doesn't inherently aim to maximize the canonical correlations by manipulating its inputs.
> > > > > > > > - In contrast, CorInfoMax networks strive to optimize the CMIs between network layers, i.e., $I^{(\epsilon)}(\mathbf{r}^{(k)},\mathbf{r}^{(k+1)})$ for $k=0, \ldots, P-1$. Our multi-layered, sample-based CorInfoMax objective (as defined in Equation (9) in our paper) facilitates the evolution of hidden layer activities and the output vector $\mathbf{y}$ to maximize the flow of correlative information across consecutive layers.
> > > > > > > >
> > > > > > > > - An intriguing future extension might involve leveraging the CCA networks from [a] as subnetworks within a CMI maximization context, where the CMI objective in (9) modified based on Eq. (*) above.
> > > > > > > >
> > > > > > > > As an important note, among different correlation measures, the correlative entropy and the corresponding correlative mutual information possess useful properties of being second-order-statistics based uncertainty and  information measures (such as additivity of self information for uncorrelated  vectors). These properties are  particularly useful for obtaining clear views for interpretibility.
> > > > > > > >
> > > > > > > > Given the relevance of the correlation concept and biological networks, we'll incorporate [a] into our references. We're grateful for your guidance on this matter.
> > > > > > > >
> > > > > > > >
> > > > > > > >
> > > > > > > >  [a] Lipshutz D, Bahroun Y, Golkar S, Sengupta AM, Chklovskii DB. A biologically plausible neural network for multichannel canonical correlation analysis. Neural Computation. 2021 Aug 19;33(9):2309-52.
> > > > > > > >
> > > > > > > >  [b] Bach, Francis R., and Michael I. Jordan. "Kernel independent component analysis." Journal of machine learning research 3.Jul (2002): 1-48.

---

> > > > > > > > > ### Comment · Reviewer_K4kq · 2023-08-15
> > > > > > > > >
> > > > > > > > > (There's a small typo in (*): the product should have been a sum.)
> > > > > > > > >
> > > > > > > > > I'm confused about the claim that CCA does not maximize the mutual information for Gaussian variables. Given that the log is an increasing function, maximizing (*) should certainly find the top canonical correlation when $p = 1$ (a similar claim is made in "Canonical correlation analysis based on information theory", *Journal of multivariate analysis* (2004). (https://core.ac.uk/download/pdf/82657228.pdf). Is the claim that there is a problem when generalizing this to $p > 1$? I don't really see how this could fail.

---

> > > > > > > > > > ### Author Response · Authors · 2023-08-15
> > > > > > > > > > **Response to Reviewer Comment**
> > > > > > > > > >
> > > > > > > > > >
> > > > > > > > > > Firstly, thank you for pointing out the typo. We've corrected it in our previous response.
> > > > > > > > > >
> > > > > > > > > > We apologize for any confusion our prior communication might have caused. To provide further clarity on the topic:
> > > > > > > > > >
> > > > > > > > > > - In our earlier response, we provided the expression for the CMI $I^{(0)}(\mathbf{x},\mathbf{y})$ using cannonical correlations $\rho_1, \ldots, \rho_p$:
> > > > > > > > > > $$ I^{(0)}(\mathbf{x},\mathbf{y})=-\frac{1}{2}\sum_{i=1}^p\log(1-\rho_i^2)\hspace{0.2in} (*)$$
> > > > > > > > > >
> > > > > > > > > > - Thus, maximizing  (*) by adjusting the second order statistics of $\mathbf{x}$ and $\mathbf{y}$, i.e., ($\mathbf{R}\_{\mathbf{x}}$, $\mathbf{R}\_{\mathbf{y}}$, $\mathbf{R}\_{\mathbf{x}\mathbf{y}}$)  would correspond to correlative information maximization.
> > > > > > > > > >
> > > > > > > > > >
> > > > > > > > > >
> > > > > > > > > >
> > > > > > > > > > If we were to clarify CCA versus CorInfoMax, these are two different maximizations over different optimization variables:
> > > > > > > > > >
> > > > > > > > > > - **CCA**: With the given second-order statistics ($\mathbf{R}\_{\mathbf{x}}$, $\mathbf{R}\_{\mathbf{y}}$, $\mathbf{R}\_{\mathbf{x}\mathbf{y}}$),  the goal is to determine the canonical coordinates $\mathbf{q}=\mathbf{V}\_\mathbf{x}\mathbf{x}$ and $\mathbf{w}=\mathbf{V}\_\mathbf{y}\mathbf{y}$, and the corresponding canonical correlations through the optimization:
> > > > > > > > > > $$ \text{maximize }\_{\mathbf{V}\_\mathbf{x},\mathbf{V}\_\mathbf{y}} \text{Tr}(\mathbf{V}\_\mathbf{x}\mathbf{R}\_{\mathbf{x}\mathbf{y}}\mathbf{V}\_\mathbf{y}^T) \hspace{0.1in} (**)$$
> > > > > > > > > > $$ \text{subject to } \mathbf{V}\_\mathbf{x}\mathbf{R}\_\mathbf{x}\mathbf{V}\_\mathbf{x}^T+\mathbf{V}\_\mathbf{y}\mathbf{R}\_\mathbf{y}\mathbf{V}\_\mathbf{y}^T=\mathbf{I}.$$
> > > > > > > > > >
> > > > > > > > > >
> > > > > > > > > > Here, the optimization variables are $\mathbf{V}\_\mathbf{x}\in \mathbb{R}^{p \times m}$ and $\mathbf{V}\_\mathbf{y}\in \mathbb{R}^{p \times n}$. The second order statistics ($\mathbf{R}\_{\mathbf{x}}$, $\mathbf{R}\_{\mathbf{y}}$, $\mathbf{R}\_{\mathbf{x}\mathbf{y}}$) are not optimization variables, but they serve as input parameters to this optimization.
> > > > > > > > > >
> > > > > > > > > > - **Correlative Information Maximization**: This involves maximizing $I^{(\epsilon)}(\mathbf{x},\mathbf{y})$ by adjusting the second order statistics  ($\mathbf{R}\_{\mathbf{x}}$, $\mathbf{R}\_{\mathbf{y}}$, $\mathbf{R}\_{\mathbf{x}\mathbf{y}}$) subject to constraints on $\mathbf{x},\mathbf{y}$ (such as the set membership constraints)
> > > > > > > > > >
> > > > > > > > > > In summary, CCA optimization variables are projection matrices, $\mathbf{V}\_\mathbf{x}\in \mathbb{R}^{p \times m}$ and $\mathbf{V}\_\mathbf{y}\in \mathbb{R}^{p \times n}$, while for CorInfoMax optimization variables are the second order statistics ($\mathbf{R}\_{\mathbf{x}}$, $\mathbf{R}\_{\mathbf{y}}$, $\mathbf{R}\_{\mathbf{x}\mathbf{y}}$)
> > > > > > > > > >
> > > > > > > > > >
> > > > > > > > > > - Our claim in our previous response was that the biological CCA  approach in reference [a]  achieves the solution for the online version of the optimization (**). It doesn't alter the input statistics ($\mathbf{R}\_{\mathbf{x}}$, $\mathbf{R}\_{\mathbf{y}}$, $\mathbf{R}\_{\mathbf{x}\mathbf{y}}$).
> > > > > > > > > >
> > > > > > > > > >
> > > > > > > > > > - In other words, the CCA network in [a] implements this maximization over the projection matrices to find canonical coordinates/canonical correlations corresponding to the provided input statistics. However, there is no maximization of the corresponding CMI by manipulating the inputs (and their statistics ($\mathbf{R}\_{\mathbf{x}}$, $\mathbf{R}\_{\mathbf{y}}$, $\mathbf{R}\_{\mathbf{x}\mathbf{y}}$)).
> > > > > > > > > >
> > > > > > > > > >
> > > > > > > > > > - In the correlative information maximization process, we actually manipulate the inputs $\mathbf{x}$ and $\mathbf{y}$ (  which are  layer activations $\mathbf{r}^{(k)}$ and $\mathbf{r^{(k+1)}}$ in our framework)  to manipulate their statistics to maximize the CMI $I^{(\epsilon)}(\mathbf{x},\mathbf{y})$, under the set membership constraints on layer activations.
> > > > > > > > > >
> > > > > > > > > >
> > > > > > > > > > - As we proposed in our previous response, a CorInfoMax extension could involve CCA networks extracting the canonical correlations vector $\rho$  for current statistics, while an outer optimization maximizes (*) by adjusting the statistics.
> > > > > > > > > >
> > > > > > > > > > We would be happy to provide further clarifications if needed.
> > > > > > > > > >
> > > > > > > > > >
> > > > > > > > > >  [a] Lipshutz D, Bahroun Y, Golkar S, Sengupta AM, Chklovskii DB. A biologically plausible neural network for multichannel canonical correlation analysis. Neural Computation. 2021 Aug 19;33(9):2309-52.

---

> > > > > > > > > > > ### Comment · Reviewer_K4kq · 2023-08-18
> > > > > > > > > > >
> > > > > > > > > > > Thanks for the further clarifications!

---

> > > > > > > > > > > > ### Author Response · Authors · 2023-08-18
> > > > > > > > > > > > **Response to the Reviewer Comment**
> > > > > > > > > > > >
> > > > > > > > > > > > We would like to thank you once again for your valuable insights and thoughtful inquiries. Your considerate observations have helped us to enhance and clarify our methodology, and we are sincerely grateful for your overall feedback.

---

### Official Review · Reviewer_MCJo · 2023-07-07

**Soundness:** 3 good
**Presentation:** 2 fair
**Contribution:** 3 good
**Rating:** 6
**Confidence:** 4

**Summary:**

The paper proposes Correlative Information Maximization as an underlying objective for biologically plausible learning. The objective produces a multi-compartmental neuron model, and can operate with feedback connection that are plastic, but not tied to the feedforward ones.

**Strengths:**

The (approximation to the) CorInfoMax objective produces a tractable model of a neuron with several compartments. This resonates with previous ideas of credit assignment with apical dendrites, and (I guess) generates experimentally testable predictions due to the specific interactions between compartments and weights.

The approach to weight symmetry is interesting and might implicitly lead to weight symmetry (although see weaknesses).

Overall, this is a novel idea, even though it is very related to previous works that use apical dendrites/predicting coding/etc. as a mechanism for credit assignment.

**Weaknesses:**

The experiments in Tab. 1 have multiple problems. There's no comparison to backprop and no explanation of used architectures in the main text. Presumably the architectures were pretty small, given poor CIFAR10 performance.

Related, all experiments show feedback alignment-level performance (i.e. good on MNIST, OK on CIFAR10 for a small network that reaches about 50% accuracy). Thus, we can’t draw any conclusions about the effectiveness of this approach without considering at least larger networks and maybe harder datasets (as feedback alignment doesn't scale beyond those cases). The minimum aim would be to train a standard ResNet18 on CIFAR10 with backprop (should be around 90% accuracy), and compare it to all algorithms in Tab. 1.

The authors also missed an important previous work -- Deep Learning without Weight Transport by Akrout et al. (2019). That paper proposes a simple mechanism for the weight transport problem that is a bit different from the one here, but it is still worth discussing in the context of backprop approximations/alternatives.

**Questions:**

Main question: are the feedback weights different from the feedforward at the end of training? I think it is an interesting question in itself, since "no" would mean your algorithm approximates backprop (similar to predictive coding approaches?). If they are different, I wonder if that could hurt performance on harder tasks and if that could be fixed somehow (akin to weight symmetry in Akrout et al.?)

Is CorInfoMax here similar to the Information Bottleneck ideas?

The abstract says
> The backpropagation algorithm... it remains an open question  whether the brain employs supervised learning mechanisms akin to it

Backprop doesn’t imply supervised learning (see VAEs, self-supervised methods and so on). The overall claim is fine, but it shouldn't be about supervised learning.

Eq. 27: should the second $e^{k+1}$ be $e^k$?

**Overall**, it is an interesting contribution but the experiments are very small-scale and the proposed approach is not compared to the main competitor, backpropagation.

----------
**Post-rebuttal**: feedback alignment-level performance is a limitation of this work, but the principled approach to derive multi-compartment models and additional evaluations done during the rebuttal justify an increase of the score from 5 to 6.


**Limitations:**

The limitations and potential impacts have been addressed.

---

> ### Author Rebuttal · Authors · 2023-08-08
>
> Thank you for your detailed review and insightful feedback. Due to strict length constraints, we had to omit some details in our responses. We eagerly look forward to providing further clarifications and answering any additional questions during the discussion period.
> > Strengths:
>
> Thank you for your positive comments.
> >..might implicitly lead to weight symmetry..
>
> Our theoretical discussion in Appendix B.2, supported by experimental angle measurements between feedback and transpose feedback weights in Appendices B.3, E.4.3, and E.5.4, indicates that CorInfoMax networks do not lead to implicit weight symmetry.
> > Weaknesses:
> >..no comparison to backprop..
>
> In the revised article, we've updated Table 1 with backpropagation and feedback alignment results, available in the rebuttal PDF.
> >.. no explanation of used architectures..
>
> The revision incorporates architecture information into the main text. Our initial submission's appendix detailed hyperparameters and network structures for each experiment. We used 2- or 3-layer fully connected networks with hidden sizes of $500$ for MNIST and $1000$ for CIFAR10 in 2-layer networks, and $500$-$500$ for MNIST and $1000$-$500$ for CIFAR10 in 3-layer networks.
> >...to train a standard ResNet18 ...
>
> Our key objective is to propose a normative learning rule leading to segregated pyramidal neuron models, addressing the weight transport problem. We focused on fully connected neural networks to establish our theoretical foundations. In our new experiments, we've achieved similar performance to biologically plausible benchmarks and BP-trained fully connected networks. We deliberately excluded CNNs, including ResNets, due to their weight-sharing feature that doesn't align with locally connected neuronal models in the brain.
> >... also missed an important previous work -- Deep Learning without Weight Transport by Akrout...
>
> Thanks a lot for pointing this relevant reference. We inserted the following change to Section 1.1.2: *"For example, the feedback
> alignment approach, which fixes randomly initialized feedback weights and adapts feedforward weights, was offered as a plausible solution [17]. Later Akrout et.al. [18 ] proposed its extension by updating feedback weights towards to the transpose of the feedforward weights"*
> >Questions:
> >..are the feedback weights different from the feedforward at the end of training?..  If they are different, I wonder if that could hurt performance..
>
> We appreciate the opportunity to clarify our model's distinctiveness. As discussed in Appendices B.2, B.3, E.4.3, and E.5.4, our model's feedback weights are not transposes of the feedforward weights, unlike conventional backpropagation networks. As discussed in Appendix B.2 of our initial submission, we can write the forward and backward predictor weights as
> $$\mathbf{W}\_{ff,\*}^{(k)}=\mathbf{R}\_{\mathbf{r}^{(k+1)}\mathbf{r}^{(k)}}(\mathbf{R}\_{\mathbf{r}^{(k)}}+\epsilon_k \mathbf{I})^{-1},$$
> $$\mathbf{W}\_{fb,\*}^{(k)}=\mathbf{R}\_{\mathbf{r}^{(k)}\mathbf{r}^{(k+1)}}(\mathbf{R}\_{\mathbf{r}^{(k+1)}}+\epsilon_k \mathbf{I})^{-1}$$
> Inspecting these, they do not just involve $\mathbf{R}\_{\mathbf{r}^{(k+1)}\mathbf{r}^{(k)}}$ and $\mathbf{R}\_{\mathbf{r}^{(k)}\mathbf{r}^{(k+1)}}$ which are transposes of each other, but also the inverse autocorrelation matrices $(\mathbf{R}\_{\mathbf{r}\^{(k)}}+\epsilon_k \mathbf{I})^{-1}$ and $(\mathbf{R}_{\mathbf{r}^{(k+1)}}+\epsilon_k \mathbf{I})^{-1}$. Consequently, the condition $\mathbf{W}\_{ff}^{(k)}={\mathbf{W}\_{fb}^{(k)}}^T$ does not generally hold true. Symmetry might be anticipated for diagonal autocorrelation matrices.
>
> In the feedforward network based standard backpropagation setting, the output of the feedback network does not directly influence the feedforward network's output, instead it generates credit signals for updating the feedforward weights. For output mean square error minimization, the feedback weights should be the transposes of the feedforward weights.
>
> However, CorInfoMax networks operate differently. Being recurrent network with feedback, these networks' dynamics—and thus their intermediate and output signals—are directly influenced by the feedback weights. Our use of equilibrium propagation-based learning ensures that the weights adapt to minimize the mean square error loss function, guided by the CorInfoMax objective. Consequently, there's no requirement for feedback weights to mirror the feedforward weights.
> > Is CorInfoMax here similar to the Information Bottleneck ideas?
>
> While both CorInfoMax and the Information Bottleneck principle derive from information theory, their relationship isn't straightforward. Traditionally, the Information Bottleneck method aims to maximize the Shannon Mutual Information (SMI) between a hidden vector and the output label, whilst simultaneously minimizing its SMI with the input. This dual goal ensures the relevance of the hidden layer to the output while promoting compression.
>
> On the other hand, the CorInfoMax framework is designed to maximize the correlative information flow across the input, hidden layers, and output in a bidirectional fashion. CorInfoMax achieves potential compression by adopting specific domain sets, such as polytopes, for the hidden and output layers. Consequently, this leads to piecewise linear activation functions and lateral inhibition neurons at these layers.
> > ..Backprop doesn’t imply supervised learning...
>
> We agree. We focused on supervised learning, primarily due to its notational convenience and alignment with the traditional form of backpropagation. However, it's crucial to note that our framework could be feasibly extended to other unsupervised and self-supervised paradigms.
> > ... should the second $\mathbf{e}^{k+1}$ be $\mathbf{e}^k$?
>
> Thanks. We corrected this in the revision.
> > ...  the proposed approach is not compared to the main competitor, backpropagation.
>
> We included new experiments with BP. Please see rebuttal pdf for the updated Table 1.

---

> > ### Comment · Reviewer_MCJo · 2023-08-10
> > **Response to rebuttal**
> >
> > Thank you for the response! Overall, I appreciate the clarifications and additional evaluations.
> >
> > However, the additional experiments confirm my concern in the original review -- the method scales similarly to vanilla feedback alignment (FA). This is evident by similar performance, and a large performance gap between CorInfoMax/FA and backprop in Tab. 2 of the rebuttal pdf. While a performance gap is not a bad thing per se, I suspect the method will fail at hard tasks just like FA. And I also suspect it could be fixed by explicitly aligning the feedback weights with the feedforwards ones.
> >
> > Since fixing the weight transport problem was one of the main goals of the paper, I think presenting a more capable method than FA is crucial.
> >
> > > We deliberately excluded CNNs, including ResNets, due to their weight-sharing feature that doesn't align with locally connected neuronal models in the brain.
> >
> > Many papers on biologically plausible deep learning use CNNs though, so including those results might help to show that the algorithm is capable of learning hard tasks (without implying that this is how the visual stream works).
> >
> > I'm open for a discussion, but currently I remain skeptical of the method's value for the field.

---

> > > ### Author Response · Authors · 2023-08-11
> > > **Response to Reviewer MCJo's Response**
> > >
> > > Thank you once again for your thoughtful feedback and the time you’ve invested in evaluating our work.
> > >
> > > We genuinely value your concerns regarding the performance on more challenging tasks. We acknowledge that achieving high performance on hard machine learning tasks remains a collective challenge for all biologically plausible models. However, this may not be the most relevant and only evaluation metric from the perspective of explaining how brains work. In fact, matching biological reality and interpretability via the use of normative principles are  important criteria (See, e.g., [a,b,c,d,e,f] below) . Indeed, the primary contribution of our article lies in its principled approach: we introduce a normative framework grounded in information theory. Biologically plausible networks comprised of multi-compartment neurons with both recurrent and asymmetric feedforward/feedback connections naturally emerge as solutions of the optimization settings  put forward through this framework. At the same time,  our approach provides principled interpretations of lateral, feedback, and feedforward connections, as well as activation functions. We can use information theoretic lens to interpret the role of these network components in terms maintaining bidirectional information flow,  avoiding embedding space  degeneracy (through lateral connections and interneurons) while achieving compression by eliminating redundancy  (through feedback/feedback connections) and domain constraints (activation functions/interneurons). At the same time, resulting networks  achieve similar performance to existing biologically plausible benchmarks without weight reuse. We are confident that the foundational nature of this framework, when extended and combined with additional biological and normative constraints, has the potential to address more practical concerns in the field.
> > >
> > > [a] Golkar S, Lipshutz D, Bahroun Y, Sengupta A, Chklovskii D. A simple normative network approximates local non-Hebbian learning in the cortex. Advances in neural information processing systems. 2020;33:7283-95.
> > >
> > > [b] Meulemans A, Zucchet N, Kobayashi S, Von Oswald J, Sacramento J. The least-control principle for local learning at equilibrium. Advances in Neural Information Processing Systems. 2022 Dec 6;35:33603-17.
> > >
> > > [c] Alonso N, Millidge B, Krichmar J, Neftci EO. A theoretical framework for inference learning. Advances in Neural Information Processing Systems. 2022 Dec 6;35:37335-48.
> > >
> > > [d] Song, Yuhang, et al. “Can the brain do backpropagation?---exact implementation of backpropagation in predictive coding networks.” Advances in neural information processing systems 33 (2020): 22566-22579.
> > >
> > >
> > > [e] Bredenberg C.,  Savin C. Desiderata for normative models of synaptic plasticity, arXiv:2308.04988, 2023.
> > >
> > >
> > > [f] Lipshutz D, Bahroun Y, Golkar S, Sengupta AM, Chklovskii DB. A normative framework for deriving neural networks with multi-compartmental neurons and non-Hebbian plasticity. PRX Life 2023.

---

> > > > ### Comment · Reviewer_MCJo · 2023-08-14
> > > > **Response to authors; score increased to 6**
> > > >
> > > > Thank you for the response!
> > > > > We are confident that the foundational nature of this framework, when extended and combined with additional biological and normative constraints, has the potential to address more practical concerns in the field.
> > > >
> > > > I ultimately agree with your points, especially the ones concerning a derivation for multi-compartment models. I still think FA-level performance is a limitation of this work, but given the discussion here and with other reviewers, I think the work is interesting enough and increase the score from 5 to 6.

---

> > > > > ### Author Response · Authors · 2023-08-15
> > > > > **Response to Reviewer Comment**
> > > > >
> > > > > We would like to thank you once again for the valuable feedback and interesting questions. Your raised concerns have been instrumental in enhancing the clarity of our paper, and we deeply appreciate your positive assessment.

---

### Author Rebuttal · Authors · 2023-08-08

We are grateful to all reviewers for their comprehensive evaluations and insightful feedback. This response covers main points and shared concerns.  **Owing to the strict length constraint**, we've made every effort to respond to individual comments and questions here and within each reviewer's rebuttal. **We will be more than happy to provide more details during the discussion phase.**

Our article offers a method grounded in information theory for the development of biologically plausible neural networks. Enhancing the clarity and accessibility of the theoretical content has been identified as an area for improvement. In addition, there have been requests for clearer derivations, explanations, and additional experiments. The revision we prepared  addresses these aspects, as highlighted below:

1. Revise Section 2.2 for better readability by *moving some details to the appendix and eliminating some unnecessary expressions*: a) Move linear approximation of correlative entropy (based on Taylor series) in (10)-(11) to a new appendix section, b) transfer CorInfoMax objective function gradient derivation details in (12)-(14) to a new appendix, c) add a new appendix section for the derivation of network dynamics equations (15)-(17), d) eliminate cross correlation matrix and detailed definition of error correlation matrix after (2).

2. Revise Section 2.2 for *better explanations and clarifications*, e.g.,

i. Replace the description after (2), which contains the definition of correlative mutual information, for clarification: "$$\overset{\rightarrow}{{I}^{(\epsilon_k)}}(\mathbf{r}^{(k)}, \mathbf{r}^{(k+1)}) =  \frac{1}{2} \log \det \left(\mathbf{R}_{\mathbf{r}^{(k+1)}} + \epsilon_k \mathbf{I}\right) - \frac{1}{2} \log \det \left(\mathbf{R}\_{\overset{\rightarrow}{\mathbf{e}^{(k+1)}\_\*}} + \epsilon_k \mathbf{I}\right) \quad (2)$$
     is the correlative mutual information between layers $\mathbf{r}^{(k)}$ and $\mathbf{r}^{(k+1)}$, $\mathbf{R}\_{\mathbf{r}\^{(k+1)}}=E(\mathbf{r}^{(k+1)}{\mathbf{r}^{(k+1)}}^T)$ is the autocorrelation matrix corresponding to the layer $\mathbf{r}^{(k+1)}$ activations,  and
$\mathbf{R}\_{\overset{\rightarrow}{\mathbf{e}^{(k+1)}\_{\*}}}$ corresponds to the error autocorrelation matrix for the best linear regularized minimum MSE predictor of $\mathbf{r}^{(k+1)}$ from $\mathbf{r}^{(k)}$. Therefore, the mutual information objective in (2) makes a referral to the regularized **forward** prediction problem represented by the optimization ...* "

ii. Provide a intuitive description for the maximization of CMI in (2): *"If we interpret the maximization of CMI in (2): the first term on the right side of (2), i.e., the correlative entropy of $(k+1)^{\text{th}}$ layer's activation vector, encourages the spread of $\mathbf{r}^{(k+1)}$ in its presumed domain $\mathcal{P}^{(k+1)}$, while the second term, i.e., the correlative entropy of forward prediction error, incites the minimization of redundancy in $\mathbf{r}^{(k+1)}$  beyond its component predictable from $\mathbf{r}^{(k)}$."*

3. Provide **a new appendix section on the role of $\epsilon$ parameter**. To summarize  a) $\epsilon$ sets a finite lower bound for the correlative entropy, b) $\epsilon$ addresses numerical optimization issues since the derivative of the $\log\det$ function is the inverse of its argument, c) $\epsilon$ acts as a regularizer for the forward and backward prediction problems (see (3) and (5) in the main article), d) $\epsilon^{-1}$  can be viewed as an indicator of the sensitivity of the CMI to the prediction error levels. This last property can be viewed from two perspectives:
 - Inspecting the CMI expression (2) above: $\epsilon_k$ is added to eigenvalues of the correlation matrices, and by definition $\mathbf{R}_{\mathbf{r}^{(k+1)}} \succeq {\mathbf{R}}\_{\overset{\rightarrow}{\mathbf{e}\^{(k+1)}\_*}}$. With $\epsilon_k$ chosen below eigenvalues of $\mathbf{R}\_{\mathbf{r}\^{(k+1)}}$, we can assume $\mathbf{R}\_{\mathbf{r}\^{(k+1)}}+\epsilon_k \mathbf{I}\approx \mathbf{R}\_{\mathbf{r}\^{(k+1)}}$. Thus, the choice of $\epsilon_k$ essentially determine how much we can reduce the correlative prediction error entropy in (2) to maximize the CMI since reducing the eigenvalues of the prediction error correlation matrix below $\epsilon_k$ would not significantly decrease the prediction error entropy. As a result, smaller $\epsilon_k$ implies more emphasis on decreasing prediction error entropy. This is in accordance with how $\epsilon^{-1}$ acts as a conductance parameter channeling prediction errors to output computation. To underline this connection, we will add additional explanation to the discussion at the end of Section 2.3.1: "The inverse of the regularization coefficient $\epsilon_k$ is related to the conductance between soma and dendritic compartments. This is compliant with the interpretation of the $\epsilon^{-1}$ in Appendix A.2 as the sensitivity parameter that determines the contribution of the prediction errors to the CMI."
 - Alternatively, consider the approximation  of (2) with linearized prediction error entropy where $\epsilon_k^{-1}$ appears as the scale of prediction error matrix dependent term:  $$\overset{\rightarrow}{{I}^{(\epsilon_k)}}(\mathbf{r}^{(k)}, \mathbf{r}^{(k+1)}) \approx \frac{1}{2} \log \det \left(\mathbf{R}\_{\mathbf{r}\^{(k+1)}}\right)- \frac{\epsilon_k^{-1}}{2} \text{Tr} \left(\mathbf{R}\_{\overset{\rightarrow}{\mathbf{e}^{(k+1)}\_*}} \right)+const$$

4. Provide **additional numerical experiments** involving comparison with standard backpropagation and feedback alignment algorithms. Table 1 updated with these experiments are provided in the PDF attachment, which confirm that the CorInfoMax has performance on par with available benchmarks.

5. Provide a section on the **limitations** of the proposed framework including hyperparameter sensitivity, contrastive optimization, and training time of our method.

---

### Decision · Program_Chairs · 2023-09-21

**Decision:**

Accept (poster)

**Comment:**

This paper examines the question of "biologically plausible learning", i.e. how can we develop learning algorithms for ANNs that work as well as backpropagation but without some of the key physiological contradictions, such as weight symmetry between the forward and backward pathways. The solution presented in this paper is to use correlative information maximization, and presents two different ways for estimating this quantity, permitting the development of mutli-compartment models that side-step the weight symmetry issue.

The reviewers raised a number of concerns, principally with respect to comparison to other techniques for dealing with weight symmetry and some issues of clarity regarding the derivations and characterizations of the analysis. But, after a fairly extensive back-and-forth, 3/4 reviewers agreed that the paper was sufficiently interesting and makes a worthwhile contribution appropriate for NeurIPS. Altogether, this led to a decision of accept.